# Suppressive cancer nonstop extension mutations increase C-terminal hydrophobicity and disrupt evolutionarily conserved amino acid patterns

Avantika Ghosh[1,2], Marisa Riester[1], Jagriti Pal[1], Kadri-Ann Lainde[1], Carla Tangermann [1,2], Angela Wanninger[1,2], Ursula K. Dueren[1], Sonam Dhamija[1,2,3,4] & Sven Diederichs [1,2]✉

Nonstop extension mutations, a.k.a. stop-lost or stop-loss mutations, convert a stop codon into a sense codon resulting in translation into the 3' untranslated region until the next in-frame stop codon, thereby extending the C-terminus of a protein. In cancer, only nonstop mutations in SMAD4 have been functionally characterized, while the impact of other nonstop mutations remain unknown. Here, we exploit our pan-cancer NonStopDB dataset and test all 2335 C-terminal extensions arising from somatic nonstop mutations in cancer for their impact on protein expression. In a high-throughput screen, 56.1% of the extensions effectively reduce protein abundance. Extensions of multiple tumor suppressor genes like *PTEN*, *APC*, *B2M*, *CASP8*, *CDKN1B* and *MLH1* are effective and validated for their suppressive impact. Importantly, the effective extensions possess a higher hydrophobicity than the neutral extensions linking C-terminal hydrophobicity with protein destabilization. Analyzing the proteomes of eleven different species reveals conserved patterns of amino acid distribution in the C-terminal regions of all proteins compared to the proteomes like an enrichment of lysine and arginine and a depletion of glycine, leucine, valine and isoleucine across species and kingdoms. These evolutionary selection patterns are disrupted in the cancer-derived effective nonstop extensions.

Nonstop extension mutations, also known as stop-loss, stop-lost or readthrough mutations, entail the conversion of a stop codon into a sense codon. This alteration results in the continuation of translation into the 3' untranslated region (UTR) until encountering the subsequent in-frame stop codon, thereby elongating the protein at its carboxy terminus (C-terminus) (Fig. 1a). These mutations have been implicated in several hereditary diseases where their occurrence results in aberrant C-terminally extended proteins with a myriad of downstream effects on protein expression including protein degradation, misfolding and aggregation[1–9]. While individual studies have described a general inhibitory effect of protein translation into 3'UTRs[10–12], the specific impact of nonstop mutations in cancer remains

[1]Division of Cancer Research, Department of Thoracic Surgery, Medical Center - University of Freiburg, Faculty of Medicine, Freiburg, Germany. [2]German Cancer Consortium (DKTK), partner site Freiburg, a partnership between DKFZ and University Medical Center Freiburg, Freiburg, Germany. [3]CSIR-Institute of Genomics and Integrative Biology, New Delhi, India. [4]Academy of Scientific and Innovative Research (AcSIR), Ghaziabad, Uttar Pradesh, India. ✉e-mail: s.diederichs@dkfz.de

largely unexplored, overshadowed by functional cancer genetics on missense and nonsense point mutations. Initial studies have only described the existence of nonstop mutations in cancer[13–15], while the first functional analysis of a cancer-derived nonstop mutation characterized extensions in the tumor suppressor gene *SMAD4* occurring in colon and pancreatic carcinomas[16]. Nonstop extension mutations in *SMAD4* resulted in the translation of a hydrophobic degron sequence which led to strong proteasomal degradation of the extended protein and a loss-of-function phenotype, thus highlighting the functional relevance of nonstop mutations in cancer.

Previously, we generated a comprehensive pan-cancer database of somatic stop-loss mutations and their corresponding C-terminal extension sequences (http://NonStopDB.dkfz.de). The NonStopDB database also provides frequencies and tumor entities for these nonstop mutations. Notably, nonstop extension mutations are similarly enriched in cancer genes as missense mutations and higher than synonymous mutations[16]. Here, we perform a flow cytometry and next-generation sequencing (NGS) based high-throughput screen of all the extensions generated by somatic cancer-associated nonstop mutations curated in the NonStopDB. We demonstrate that the C-terminal extensions frequently lead to the loss of protein expression as a function of length and hydrophobicity, as in the case of stop-loss mutations in the tumor suppressor *PTEN*, that lead to reduced expression and function of the PTEN protein. Furthermore, we uncover that nonstop extension mutations disrupt the evolutionarily conserved patterns in the C-terminal amino acid composition across species and kingdoms.

## Results

### High-throughput screening platform for testing cancer-derived nonstop extensions

The NonStopDB curated mutational pan-cancer data from the Catalog of Somatic Mutations in Cancer (COSMIC; version 89)[17] for somatic nonstop mutations[16]. At the tumor entity level, the database distinguishes 96 different tumor entities annotated according to COSMIC. Most of the 3412 cases with nonstop mutations are found in tumors of the digestive system (1255) followed by the genitourinary system (736) and the respiratory system (712). Looking at the individual tumor entity, most nonstop mutations are found in the adenocarcinoma of the large intestine (569) followed by the adenocarcinoma of the lung (243) and carcinoma of the breast (201). The genes with most nonstop extension mutations are *ACO2* (13), with all mutations found in head and neck squamous cell carcinoma and *PRKCH* (11) with all mutations found in glioma. The cancer gene with most nonstop extension mutations is *SMAD4* (11) with all mutations found in colon, pancreatic and bile duct cancer, which are known to be driven by different loss-of-function mutations in *SMAD4*[16], thus, representing entity-specific nonstop extension mutations. Contrastingly, the nonstop mutations in the cancer gene *CDKN2A* (9) are found in different entities including lung squamous cell carcinoma (2), lung adenocarcinoma (2), lung small cell carcinoma (1), head and neck squamous cell carcinoma (1), esophagus squamous cell carcinoma (1), pancreas ductal adenocarcinoma (1) and malignant melanoma (1). To perform a high-throughput analysis of all nonstop extensions, we designed a library of oligonucleotides (oligos) containing the 2335 different nonstop mutation derived C-terminal extensions (NSdb_MUT) from the NonStopDB. Each oligo was designed to start with the amino acid replacing the lost stop codon and to end with one of the three stop codons (TAG, TAA or TGA). In addition, 2172 oligos were included as wildtype controls of the mutant nonstop oligos (NSdb_WT), where the sequences began and ended with a stop codon, with the intervening nucleotide sequence being identical to the corresponding NSdb_MUT sequence. The three stop codons alone were also included as negative controls. Apart from the NSdb sequences, we designed 200 artificial sequences each incorporating one of the 20 amino acids repeated from one to ten

times. As positive controls, 25 sequences previously described as C-terminal degrons (degrons) were included. The degron sequences were known to bear motifs that resulted in protein elimination by a mechanism described as "Destruction via the C-END" (DesCEND)[18]. The 25 degron sequences are regulated by various Ubiquitin ligases and the sequences included in the library represent the specific amino acid sequence motifs delineated in previous studies[18–29] (Fig. S1a).

To analyze the effect of C-terminal extensions resulting from nonstop mutations on protein expression, we constructed an eGFP-IRES-mCherry dual fluorescent reporter[10,12]. The library of extensions was fused to the C-terminus of eGFP and their impact was assessed through eGFP expression, with mCherry serving as an internal normalization control (Fig. 1b). Studies of C-terminal regulation using reporters have either been performed using individual reporters[10] or in the context of a screen, using a constant sequence length[12,29]. Since we aimed to analyze the complete length of the C-terminal extensions from the NonStopDB which were naturally variable in length, we normalized all the extensions in the library to the same size by filling in a non-translated inert sequence[30] after the extension and stop codon, for extensions shorter than 252 nucleotides (nt). The inertness of the sequence was verified by testing reporters containing wildtype or mutant extensions of *SMAD4* with or without the inert sequence. No impact on eGFP expression was observed due to the inert sequence (Fig. S1b). Extensions longer than 252 nt were tiled across multiple oligos.

We then generated a pooled library of HEK293T cells expressing eGFP fused to one of the C-terminal extension sequences from the oligo pool at its C-terminus. Using flow cytometry, we observed that the presence of the C-terminal extensions resulted in a range of eGFP expression from low to high, indicating the presence of sequences that disrupted protein expression and sequences that had no effect, respectively. Using fluorescence-activated cell sorting (FACS), we sorted the library of HEK293T cells based on eGFP expression into three populations: eGFP low, eGFP intermediate and eGFP high, and sequenced each of the three populations using NGS (Fig. 1b). We performed the screen in four biological replicates, achieving a statistically significant correlation between the replicates (Fig. 1c, Fig. S1c).

### Nonstop mutation derived C-terminal extensions frequently disrupt protein expression

As a first analysis, we determined the fraction of normalized read counts in the eGFP-reduced (eGFP low or intermediate) populations compared to all reads for the respective extension. For the NSdb_MUT sequences, a median of 75% of the reads were found in the eGFP-reduced populations similar to the positive control degron sequences with 80%. In contrast, only 21% of reads from the NSdb_WT sequences and 18% of reads from the three stop codons were found in the eGFP-reduced populations (Fig. 2a). Thus, the NSdb_MUT sequences showed higher prevalence in the eGFP-reduced populations, indicating their similarity to known degrons, while the NSdb_WT sequences and the stop codons were depleted in these populations.

To calculate an enrichment score for each extension in each replicate, we used the median of the ratios of the maximum value of the normalized reads from the eGFP-reduced populations, divided by the reads in the eGFP high population (median enrichment score, M.E.) (Fig. S1d). High enrichment scores thus indicated a strong enrichment in the eGFP-reduced populations, i.e., a negative impact of the extension on protein abundance. Comparing the medians of each group, the NSdb_MUT sequences showed a median enrichment that was approx. 16-fold greater than the median enrichment of the three stop codons and 13-fold greater than the median enrichment of the NSdb_WT sequences (Fig. 2b). The C-terminal extensions from known degrons as well as the previously described *SMAD4* extensions[16] showed a 20-fold and 19-fold greater median enrichment than the three stop codons,

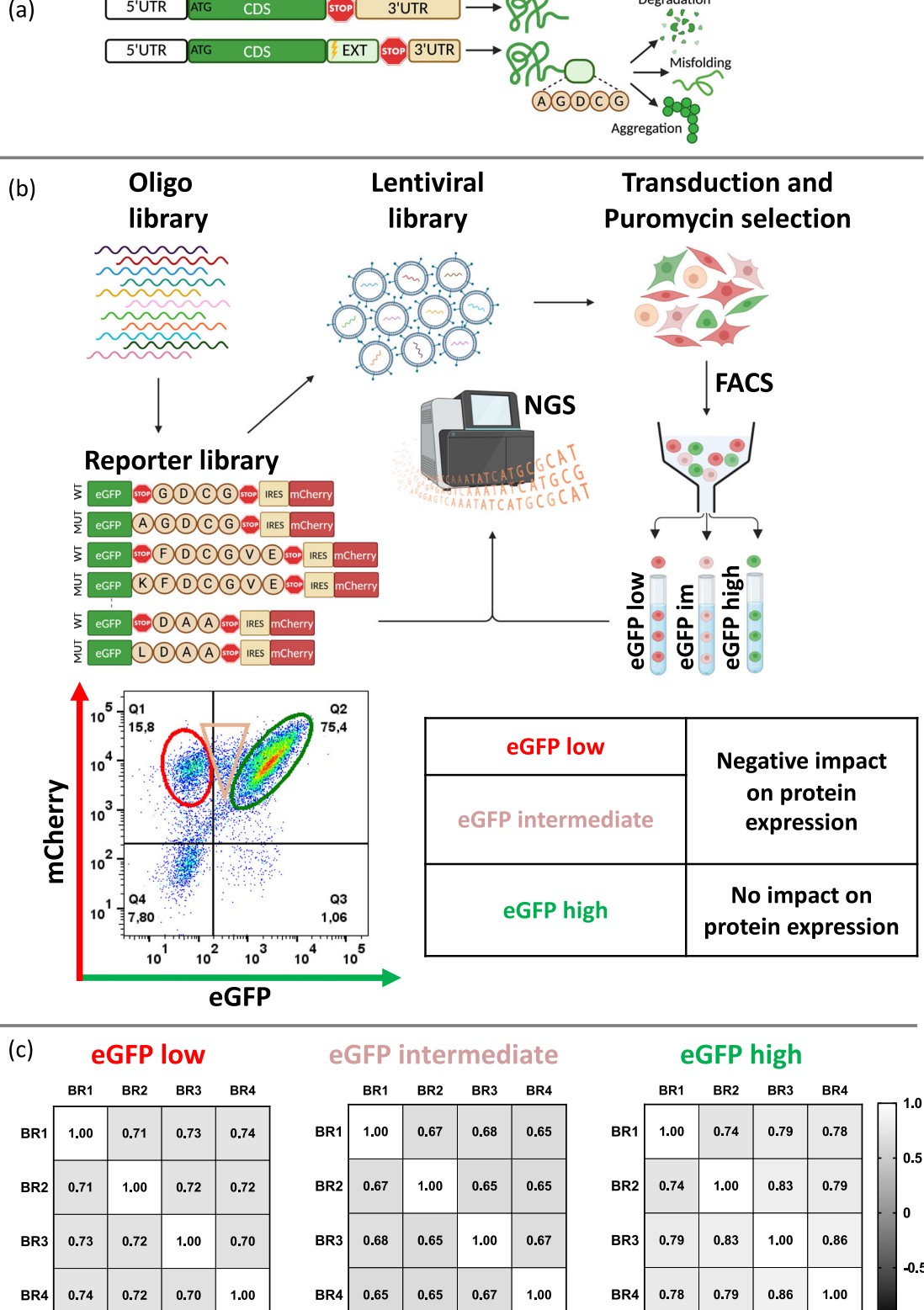

**Fig. 1 | High-throughput screening platform for testing cancer-derived nonstop extensions. a** Nonstop extension or stop-loss mutations convert a stop codon into a sense codon with various possible impacts on protein expression. CDS, coding sequence; EXT, extension; UTR, untranslated region. **b** Schematic of the high-throughput screening platform for analyzing the effect of stop-loss mutation-derived C-terminal extensions. Representative flow cytometry plot depicting the range of eGFP expression in HEK293T cells after transduction with the pooled library of fusion eGFP-nonstop extension reporters. The transduced cells were sorted by FACS into three populations reflecting different impacts on protein expression. im, intermediate. **c** Spearman correlation co-efficients were obtained for the NGS read counts in the four biological replicates for the FACS-sorted eGFP low, eGFP intermediate and eGFP high populations, respectively. The scale depicts the range of the Spearman correlation coefficient from −1.0 to 1.0. BR, biological replicate. Panels (**a** and **b**) were created in BioRender. Ghosh, A. (2024) BioRender.com/w29n909 and Ghosh, A. (2024) BioRender.com/h19s334, respectively.

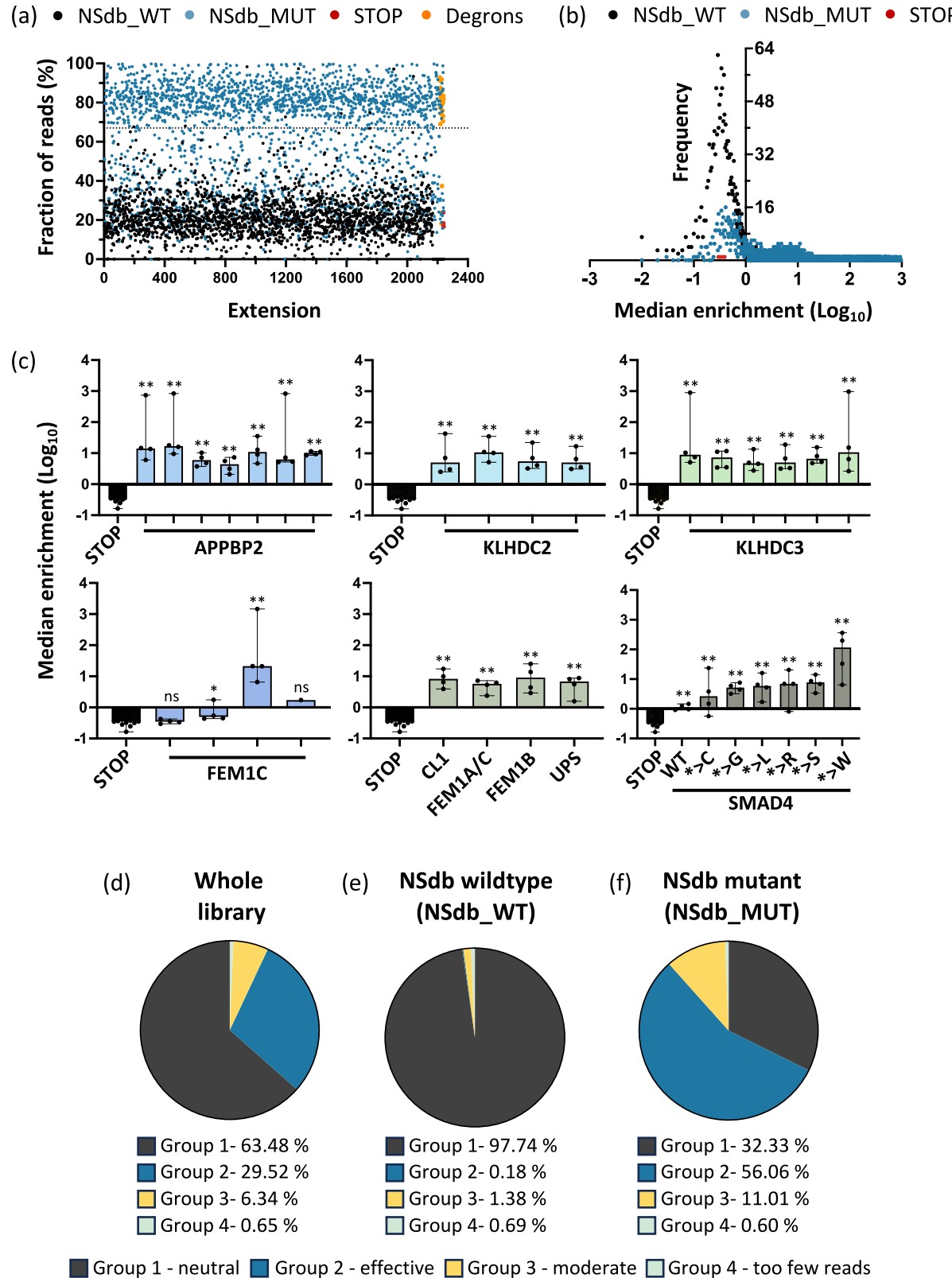

**Fig. 2 | Nonstop mutation-derived C-terminal extensions frequently disrupt protein expression. a** Median of the fraction of normalized reads in the eGFP reduced (low and intermediate) populations compared to the sum of reads in all populations for each extension from the four biological replicates. The eGFP reduced populations were enriched in stop-loss mutation derived mutant C-terminal extensions (NSdb_MUT, blue) relative to wildtype controls (NSdb_WT, black). The dotted line represents the theoretical threshold (67%) of even distribution of an extension in each of the three sorted populations. **b** NSdb_MUT sequences (blue) had higher median enrichment scores (M.E.) than the NSdb_WT extension sequences (black),

demonstrating that the presence of cancer-derived C-terminal extensions disrupts protein expression. **c** The extension sequences of the positive control degrons showed a high M.E. Error bars depict median with 95% CI. *$P < 0.05$, **$P < 0.01$ two-tailed Mann-Whitney U test, comparing each of the respective extensions independently to the three STOP controls. **d**–**f** Pie chart depicting the distribution of the extensions in the library into the four M.E. based categories for all the oligos in the library (**d**) for the NSdb_WT extensions (**e**) and the NSdb_MUT extensions (**f**). In all panels, $n = 4$ biologically independent replicates of the screen.

respectively (Fig. 2c), thus, confirming the validity of our screening results. Based on the median enrichment scores, the screening results for our library could be divided into four groups: 1 - neutral, extensions without any effect on eGFP; 2 - effective, extensions that led to decreased eGFP; 3 - moderate, extensions that showed an intermediate loss of eGFP; and 4 - too few reads, extensions with very low representation in the sequencing libraries which could not be analyzed for effectiveness (Fig. S1d). For the NonstopDB sequences, 56.1% of NSdb_MUT effectively inhibited protein expression, while only 0.2% of all NSdb_WT sequences showed this effect (Fig. 2d-f and Suppl. Data 1). In addition, we analyzed the presence of motifs known to function as C-end degrons, e.g. terminal glycine or terminal di-glutamic acid[29], in the NSdb_MUT extensions and found them to lead to a further increase to 61% effective extensions (group 2). Further, the occurrence of effective mutations was slightly higher (58.5%) for the mutations recurrently listed in COSMIC over single-entry mutations (55.7%).

A ROC (receiver operating characteristic) curve analysis comparing the M.E. of the screen negative controls (the three stop codons and the 2172 NSdb_WT extensions) to the screen positive controls (25 degrons and the six SMAD4 extension sequences) showed an AUC (area under the curve) of 0.96 (95% CI 0.92–1.00, p < 0.0001) (Fig. S1e). This demonstrated the validity of the screen M.E. scores in providing a greater than chance prediction of the effect of the C-terminal extension sequences.

These results highlighted both the functionality of our screen and demonstrated that cancer-derived nonstop mutations frequently led to C-terminal extensions that decreased protein expression.

To compare the experimental functional screening results to in silico predictions, we calculated the scores for nonstop extension mutation effects using CSCAPE[31], FATHMM-XF[32] and CADD[33] and their predictions as "oncogenic" or "benign". Notably, the group of neutral extensions (group 1) and the group of extensions with a suppressive effect (group 2) were rather equally distributed between the predictions as oncogenic or benign (Fig. S2a). Further, a ROC curve analysis comparing the group 1 versus group 2 extensions showed AUC values of only 0.53 (95% CI 0.50–56, p = 0.023), 0.54 (95% CI 0.51–0.57, p = 0.006) and 0.52 (95% CI 0.49–0.55, p = 0.17), for CSCAPE, FATHMM-XF and CADD, respectively (Fig. S2b). The almost even distribution of the neutral and effective extensions as oncogenic and benign along with AUC values only slightly higher than 0.5 indicated that these prediction tools were only of minor predictive value for the estimation of the effects of nonstop extension mutations.

## Cancer associated C-terminal extensions in tumor suppressor genes cause a loss of expression

Of the extension sequences cataloged in the NonStopDB, more tumor suppressor genes (TSGs, n = 77) than oncogenes (OGs, n = 66) possess nonstop mutations classified by the Cancer Gene Census (CGC)[34]. In line with this, our screen showed that C-terminal extensions derived from TSGs were more often effective in comparison to those derived from OGs, especially for strongly suppressive extensions (Fig. 3a, Suppl. Data 2). Multiple C-terminal extensions arising from nonstop mutations in TSGs strongly decreased protein expression (Fig. 3b, Fig. S2c). We experimentally validated this effect for ten stop-loss mutations in six prominent TSGs from the CGC Tier 1: APC, B2M, CASP8, CDKN1B, MLH1 and PTEN. These genes either had a very strong effect in the screen or showed suppressive effects of multiple independent extensions resulting from different nonstop mutations in the same gene (Fig. 3b). Taken together, these ten TSG extensions showed a median enrichment that was 13-fold greater than the M.E. of the entire library and 6.8-fold greater than the M.E. of all NSdb_MUT sequences.

To verify the effect of these extensions independent of the screen, we cloned the individual extensions downstream of eGFP in the eGFP-IRES-mCherry reporters and analyzed their impact by fluorescence microscopy (Fig. 3c) and flow cytometry (Fig. 3d). In both assays, the extensions significantly reduced eGFP fluorescence, with an average 46% reduction in the eGFP/mCherry ratio in flow cytometry (Fig. 3d,e).

To analyze the impact of the effective (group 2) TSG extension sequences linked to their own coding sequence, we cloned the C-terminal extensions with either the wildtype stop codon or the mutated stop codon for B2M, CDKN1B (* > L and *>S) and MLH1 (* > K and *>Y), downstream of their respective HA-tagged CDS. We observed a significant loss of protein expression of the HA-tagged CDS with mutated stop codons for all of these five extensions in comparison to the corresponding wildtype CDS with intact stop codons (Fig. 3f-j, Fig. S3a-e). Additionally, we analyzed the effect of neutral (group 1) nonstop extensions from three genes, ACO2, GATA1 and PRKCH, and cloned their HA-tagged CDS and extensions. Concordant with the reporter screening results, the protein expression from the CDSs of ACO2, GATA1 and PRKCH were not significantly affected by the nonstop mutations (Fig. 3k-m, Fig. S3f-h).

These data serve as a validation of the results from our screen and indicate a relevant role of nonstop mutations in suppressing these TSGs.

## Functional impact of stop-loss mutations in the tumor suppressor gene PTEN

The Phosphatase and Tensin homolog (PTEN) is a key tumor suppressor gene and is one of the most commonly mutated TSGs in human cancer[35,36] with several studies demonstrating the essential role of PTEN in multiple tissue types[37–39]. The dephosphorylation of phosphatidylinositol-3,4,5-trisphosphate (PIP3), the main substrate of PTEN, suppresses the activation of the PI3K-AKT-mTOR signaling pathway, a key pathway involved in cell growth, survival, and metabolism[40]. Thus, a loss of PTEN expression and/or function can lead to aberrant cell signaling and uninhibited growth (Fig. S3i). Given the crucial role of PTEN in cancer, we wanted to characterize and validate the functional impact of its nonstop extension mutations. To analyze the effect of C-terminal extensions on PTEN, we used both an overexpression system as well as endogenous mutagenesis to create a nonstop PTEN mutant.

The exogenous expression of the HA-tagged PTEN coding sequence (CDS) with nonstop extension mutations was significantly lower than the corresponding HA-tagged wildtype protein where translation stopped at the natural stop codon (Figs. 4a, b). Further, the loss of PTEN protein expression could be partially rescued by inhibiting proteasomal degradation with Bortezomib (Figs. 4c, d).

For the specific introduction of the nonstop point mutation endogenously into the PTEN gene, we employed CRISPR precision genome editing in the HEK293T cell line. We created two heterozygous clonal cell lines (Figs. S3j, k) bearing the STOP> Leucine (*> L) mutation, the PTEN nonstop mutation that showed the strongest effect in our screen (Fig. 3d). In these mutant clonal cell lines, the nonstop mutation in PTEN led to decreased protein expression (Figs. 4e, f). The PTEN nonstop mutation cell lines showed an increase in the levels of AKT phosphorylation at Threonine-308 (Thr308), demonstrating a functional effect of the mutation even in a heterozygous setting (Figs. 4e, g).

In summary, these exogenous and endogenous models verified the loss of PTEN protein expression and function by the PTEN C-terminal extensions caused by nonstop mutations.

## Impact of hydrophobicity on nonstop extension induced PTEN degradation

In the screen (Fig. 3b) as well as in the validation experiments (Figs. 3c–e, 4a), we noticed that the three different nonstop mutations affecting PTEN showed different strengths of suppressive effects, with *> L and *> C showing stronger effects than *> S (Fig. 3e). Due to the short size of the PTEN extension (with just eight amino acids) and only

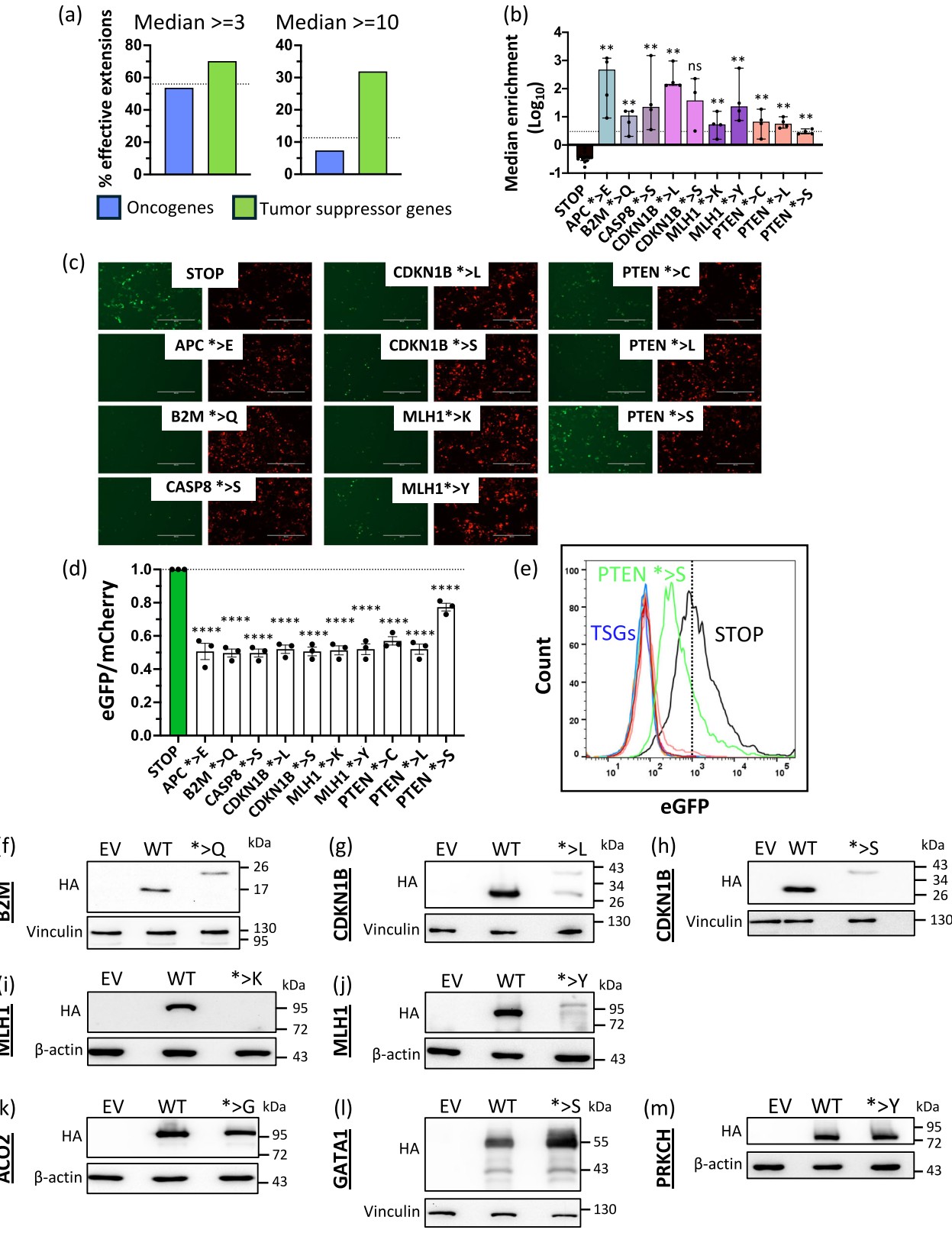

the single amino acid substitution at the stop codon differing between extensions, coupled with the established link between hydrophobicity and protein destabilization[12,16], we hypothesized that the hydrophobicity of the substituted amino acid might govern this observed variation.

To characterize this impact of C-terminal hydrophobicity on PTEN, we created 20 individual eGFP-fusion reporters for the

*PTEN* extension, each bearing one of the 20 amino acids in place of the lost stop codon. The observed reduction of the eGFP/mCherry ratio strongly correlated with the hydrophobicity of the amino acid replacing the stop codon. A hydrophilic or charged amino acid rendered it less effective while the presence of large hydrophobic amino acids increased the effectivity (Figs. 5a, b).

**Fig. 3 | Cancer-associated C-terminal extensions in tumor suppressor genes cause a loss of expression. a** Extensions of tumor suppressor genes (TSGs) showed a greater enrichment in comparison to oncogenes (OGs) in the screen for M.E. > = 3 and M.E. > = 10, demonstrating a stronger loss of eGFP expression in TSG-associated extensions. **b** The six TSGs selected for validation had high M.E.s demonstrating their strong suppressive effect on eGFP expression. *n* = 4 biologically independent replicates of the screen. Error bars depict median with 95% CI. ***P* < 0.01, two-tailed Mann-Whitney U test, comparing each of the respective extensions independently to the three STOP controls. **c-e**, eGFP fusion reporters containing TSG-derived C-terminal extensions showed a loss of eGFP expression both in fluorescence microscopy (**c**) and flow cytometry (**d, e**). *n* = 3 biologically independent replicates. Error bars depict mean ± SEM. *****P* < 0.0001, one-way ANOVA and Fisher's LSD test, comparing each of the respective extensions

independently to the control extension containing only the stop codon TGA (STOP). **f–j** Effective nonstop extension mutations (group 2) fused to the HA-tagged CDS of TSGs *B2M* (**f**), *CDKN1B* (**g, h**) and *MLH1* (**i, j**) lead to a loss of protein expression in western blot analysis in HEK293T cells after transient transfection. The wells on either side of the wildtype were intentionally left blank, except in (**h**) where only the well on the right of the WT was left blank. EV, empty vector; WT, wildtype. **k–m** Neutral nonstop extension mutations (group 1) fused to the HA-tagged CDS of the genes *ACO2* (**k**), *GATA1* (**l**) and *PRKCH* (**m**) do not have an impact on protein expression in western blot analysis in HEK293T cells after transient transfection. The wells on either side of the wildtype were intentionally left blank. EV, empty vector; WT, wildtype. The data presented in panels **f–m** are representative of three independent biological replicates. The quantification is provided in Fig. S3a-h and source data are provided in the source data file.

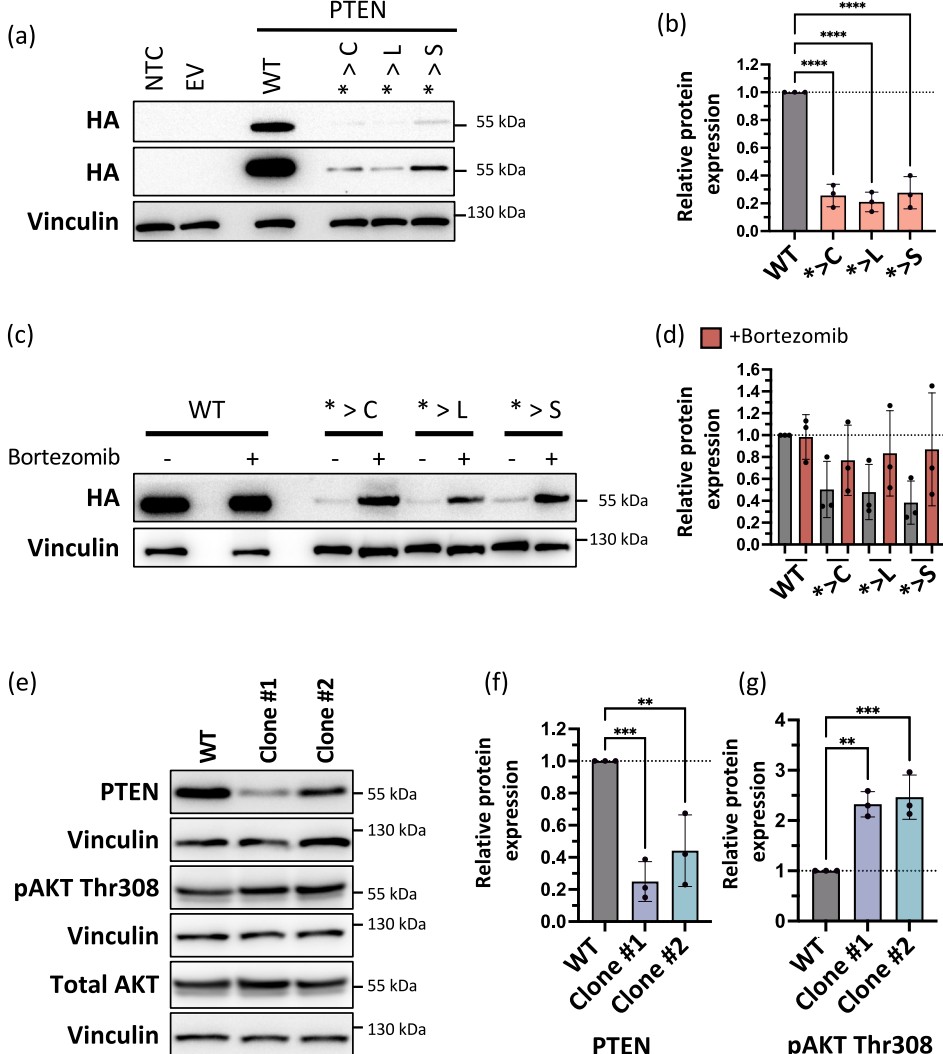

**Fig. 4 | Functional impact of nonstop mutations in the tumor suppressor gene *PTEN*. a**, **b** Nonstop extension mutations in the *PTEN* CDS led to a loss of protein expression in western blot analysis in HEK293T cells after transient transfection (**a**, **b**). The two HA panels in (**a**) show the expression of HA-tagged PTEN under short (top) or long (bottom) exposures. the NTC, no transfection control; EV, empty vector; WT, wildtype. The wells on either side of the wildtype were intentionally left blank. Relative protein expression was normalized to Vinculin (**b**). *****P* < 0.0001, one-way ANOVA and Fisher's LSD test comparing each of the three respective extensions to the WT. **c**, **d** Expression of the HA-tagged *PTEN* nonstop extension mutants after transient transfection into HEK293T cells was partially rescued by

0.1 µM Bortezomib treatment for 20 h as determined by western blot analysis. The wells on either side of the wildtype were intentionally left blank. Relative protein expression was normalized to Vinculin (**d**). **e–g** CRISPR-Cas12 generated heterozygous endogenous *>L mutant HEK293T cells showed a lower *PTEN* expression (**e**, **f**) and an upregulation at phospho-AKT Thr308 (**e**, **g**). WT, wildtype. Relative protein expression was normalized to Vinculin (**f**, **g**). ****P* < 0.001, ***P* < 0.01, one-way ANOVA and Fisher's LSD test comparing each of the clones to the WT. In all panels, *n* = 3 biologically independent replicates were carried out and error bars depict mean ± standard deviation. Source data are provided in the source data file.

To further assess the effect of hydrophobicity in the PTEN extension, we used our PTEN fusion reporters containing the *> C and *> L extensions and altered the extension by mutating the hydrophobic residue isoleucine present after the mutated stop codon to the hydrophilic charged residue glutamic acid. Indeed, interrupting the stretch of hydrophobic amino acids rescued protein expression (Fig. 5c,d).

### Hydrophobicity is a general key determinant of nonstop extension mediated loss of expression

Next, we aimed to analyze whether there was a general correlation between hydrophobicity of cancer-derived nonstop extensions and their suppressive effect. First, we compared the length of the neutral extensions and the effective extensions and found the latter to be 2.6-fold longer than the former (Fig. 5e). We globally analyzed hydrophobicity between the neutral and the effective extensions using two independent scales for hydrophobicity (Miyazawa and Kyte-Doolittle scales)[41,42] to strengthen the robustness of our analysis. Independent of the hydrophobicity scale used, the effective extensions possessed a significantly greater hydrophobicity than the neutral extensions (Figs. 5f, g). We also observed that the extensions occurring in tumor suppressor genes exhibited greater hydrophobicity than those in oncogenes (Figs. S4a, b). Additionally, we utilized the 200 artificial extensions comprising one to ten copies of each of the 20 proteinogenic amino acids included in the screen. On average (Fig. 5h) and for individual amino acids (Fig. S4c), amino acids possessing higher hydrophobicity effectively inhibited protein expression. In contrast, amino acids with low hydrophobicity were generally neutral. For the extensions possessing between 1 and 10 hydrophobic amino acids, the longer the stretch of hydrophobic amino acids, the higher was the likelihood of eliciting a loss of expression compared to extensions with an equivalent number of hydrophilic amino acids (Fig. S4c). When globally analyzing all extension-derived newly created C-termini in the screen in comparison to all wildtype C-termini of the respective coding sequences, only the effective C-terminal extensions (group 2), but not the neutral extensions (group 1), had a significantly increased hydrophobicity (Figs. 5i–l). This observation was also confirmed in the pairwise comparison of wildtype protein coding sequences versus the respective extension-derived C-termini in the screen, with a larger difference in hydrophobicity for the effective over the neutral extensions (Figs. S4d, e).

To further assess the impact of hydrophobicity on the suppressive effect of the extensions, we compared, in total, 28 different parameters for their correlation to the median enrichment in the screen (Figs. S5a–h). For the hydrophobicity based on the Miyazawa scale or on the Kyte-Doolittle scale as well as for the extension length, the Spearman correlation to the enrichment in the functional screen was high and strongly significant (Figs. S5a, b). In contrast, the correlation coefficients for the twenty individual amino acids were all much lower (Fig. S5c). Comparing the median enrichment between extensions with components of known degrons like a C-terminal glycine (G) or an arginine at the minus three position (R at −3) showed only weaker differences and for a C-terminal di-glutamic acid (EE), a C-terminal glutamine (Q) or a valine at minus two position (V at −2), there were no significant differences in enrichment (Figs. S5d–h). Thus, out of these tested parameters, the hydrophobicity and length of the extension showed the strongest correlation with the median enrichment in the screen for suppressive extensions.

### The amino acid composition at the C-termini of proteins differs from the proteome across species and kingdoms and is altered by cancer-associated nonstop extension mutations

To further investigate the parameters determining the suppressive effect of extensions at the C-terminus, we analyzed the amino acid distribution at the C-termini (Fig. 6a) of the effective extensions (group

2) compared to either the neutral extensions (group 1, Fig. 6b) or the human proteome (Fig. 6c). This comparative view of the amino acids at the last 10 positions in the extension revealed a strong enrichment of four (phenylalanine (F), isoleucine (I), leucine (L) and cysteine (C)) of the five most hydrophobic amino acids (methionine (M) presenting an exception) in the effective extensions versus the neutral extensions or the proteome. Conversely, the three most hydrophilic amino acids, lysine (K), aspartic acid (D), and glutamic acid (E), were depleted in the effective extensions at the last 10 positions. Glycine (G) was strongly enriched only at the last position in the effective group 2, both in comparison to the neutral group 1 and the proteome, in accordance with its known role as a C-end degron[18,29] (Figs. 6b, c). Thus, the enrichment of hydrophobic amino acids and the depletion of hydrophilic amino acids are general key deterministic features of the C-termini of effective nonstop extensions. However, hydrophobicity does not represent the only determinant as evidenced by the patterns observed for residues like methionine and glycine.

To globally assess and compare the amino acid distribution at the C-terminus, we analyzed the proteomes of eleven different species across all kingdoms of life. For this global analysis, we quantified the amino acid composition at the last ten amino acids of the proteins in the eleven proteomes in comparison to the amino acid composition of the respective entire proteomes. As previously discussed for the C-end rule, we found notable and highly conserved evolutionary patterns. In most species, the C-termini were generally depleted of glycine, valine, leucine, isoleucine, and alanine, while lysine and arginine were overrepresented (Fig. 6d). Thus, as in the human proteome, hydrophobic residues (like valine, leucine, isoleucine) were generally depleted at the C-terminus across all species, while hydrophilic amino acids (like lysine, arginine) were enriched. Here too, other determinants seemed to impact the C-terminal amino acid distribution. Glycine, despite its lower hydrophobicity was strongly depleted, while phenylalanine with a high hydrophobicity was not depleted. Conversely, the hydrophilic, negatively charged residues aspartic and glutamic acid were not enriched compared to the positively charged lysine and arginine (Fig. 6d).

In comparison to the evolutionarily selected patterns of C-terminal amino acid composition in proteomes, the C-terminal extensions of the NSdb_MUT showed an inverse pattern for a number of amino acids. In contrast to the C-termini of proteomes across species, the effective nonstop extensions showed an enrichment of hydrophobic residues leucine or isoleucine. Conversely, effective nonstop extensions were depleted in lysine and arginine, which were enriched at the C-termini of all eleven species compared to their proteomes (Fig. 6e).

Lastly, glycine at the terminal position is known to destabilize proteins[18,29]. The presence of glycine was strongly depleted at the −1 C-terminal position compared to the nine preceding amino acids in all eukaryotic species, but not in *E. coli*, due to the differences in the ubiquitin-proteasome systems between prokaryotes and eukaryotes[29,43,44] (Fig. 6f). In the human proteome, the frequency of glycine was reduced from 6.1% in the penultimate nine amino acids to 3.4% at the terminal position. While this remained unchanged for neutral extensions (group 1: 2.9%), the frequency for C-terminal glycine was as high as 7.5% for cancer-derived effective nonstop extensions (Fig. 6g), concordant with the significantly increased median enrichment seen for extensions with a C-terminal glycine (Fig. S5d).

In summary, these analyses highlight the loss of evolutionarily conserved amino acid signatures that distinguish the C-termini from the rest of the proteomes by cancer-derived effective nonstop extensions thereby leading to a decrease in protein abundance.

## Discussion

In the age of genomics and precision oncology, there is a great need for characterizing the impact of all types of mutations in cancer. Despite the moderate frequency of nonstop extension mutations, their

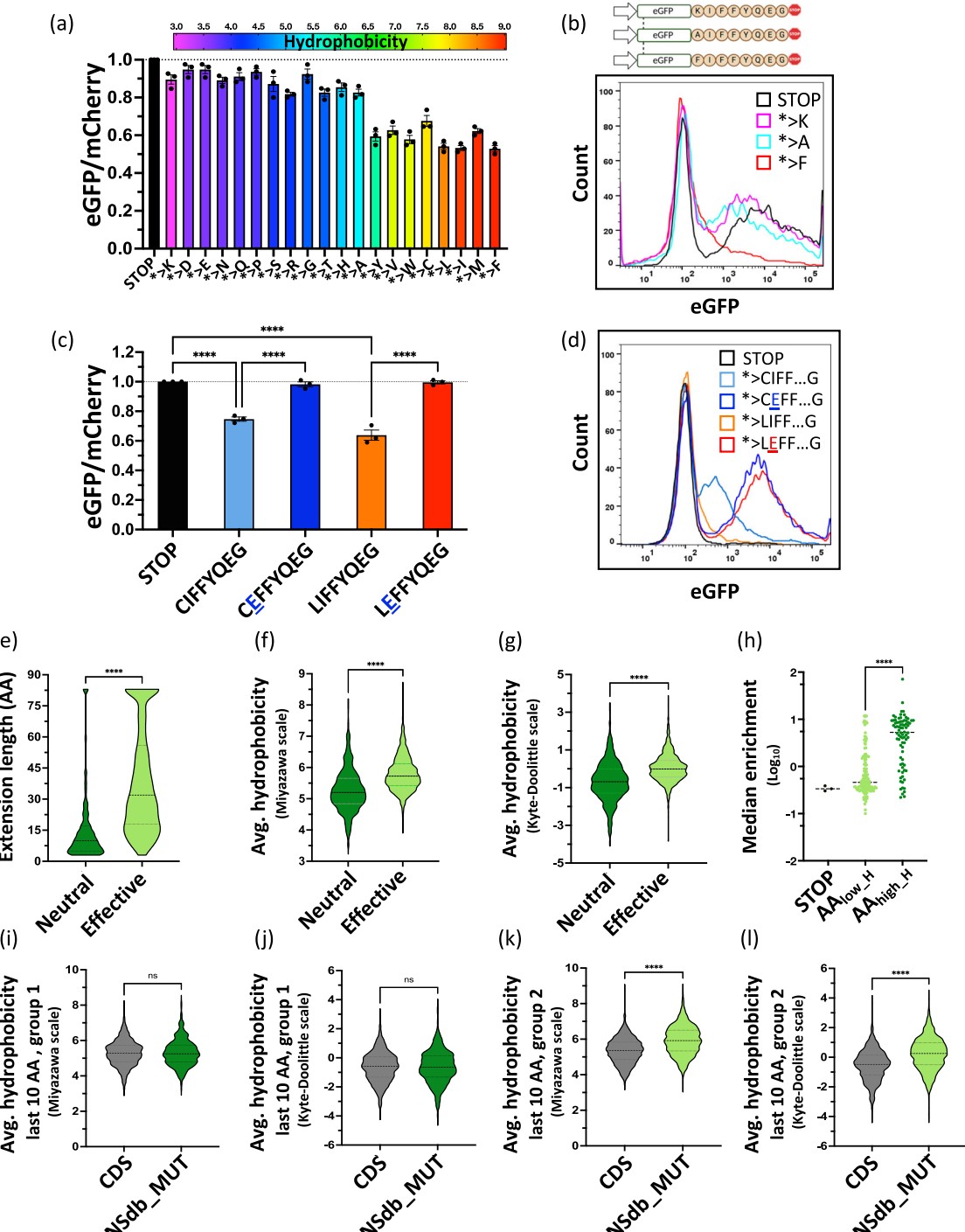

**Fig. 5 | Hydrophobicity is one determinant of nonstop extension-mediated loss of expression. a**, **b** Hydrophobicity of the amino acid replacing the STOP codon is a determinant of protein loss in the eGFP-fusion *PTEN* extension reporters. The STOP codon was mutated to each of the 20 proteinogenic amino acids showing a positive correlation of hydrophobicity with the loss of protein expression. $n = 3$ independent biological replicates. Error bars depict mean ± SEM. Panel b was created in BioRender. Ghosh, A. (2024) BioRender.com/u94b137. **c**, **d**, The effect of hydrophobicity on the loss of eGFP was rescued by interrupting a stretch of hydrophobic amino acids with a hydrophilic, charged amino acid in the *> C and *> L eGFP fusion reporters with the *PTEN* extension evidenced by flow cytometry. $n = 3$ independent biological replicates. Error bars depict mean ± SEM. ****$P < 0.0001$, one-way ANOVA and Fisher's LSD test. **e**–**g** Effective (group 2) cancer-derived nonstop extensions

were significantly longer (**e**) and more hydrophobic than neutral extensions (group 1) using both the Miyazawa (**f**) and the Kyte-Doolittle (**g**) scales for hydrophobicity. ****$P < 0.0001$, two-tailed Mann-Whitney U test. **h** In the fusion reporters containing artificial sequences, the more hydrophobic amino acids elicited a stronger loss of eGFP in comparison to hydrophilic amino acids. ****$P < 0.0001$, two-tailed Mann-Whitney U test. **i**–**l** The C-termini (last 10 amino acids) of effective (group 2) extensions possess a significantly higher hydrophobicity than the C-termini (last 10 amino acids) of the wildtype protein coding genes using both the Miyazawa (**k**) or the Kyte-Doolittle (**l**) scales for hydrophobicity. The C-termini of neutral extensions do not differ in hydrophobicity from the CDS of wildtype proteins (**i**, **j**). ****$P < 0.0001$, ns $P > 0.05$, two-tailed Mann-Whitney U test.

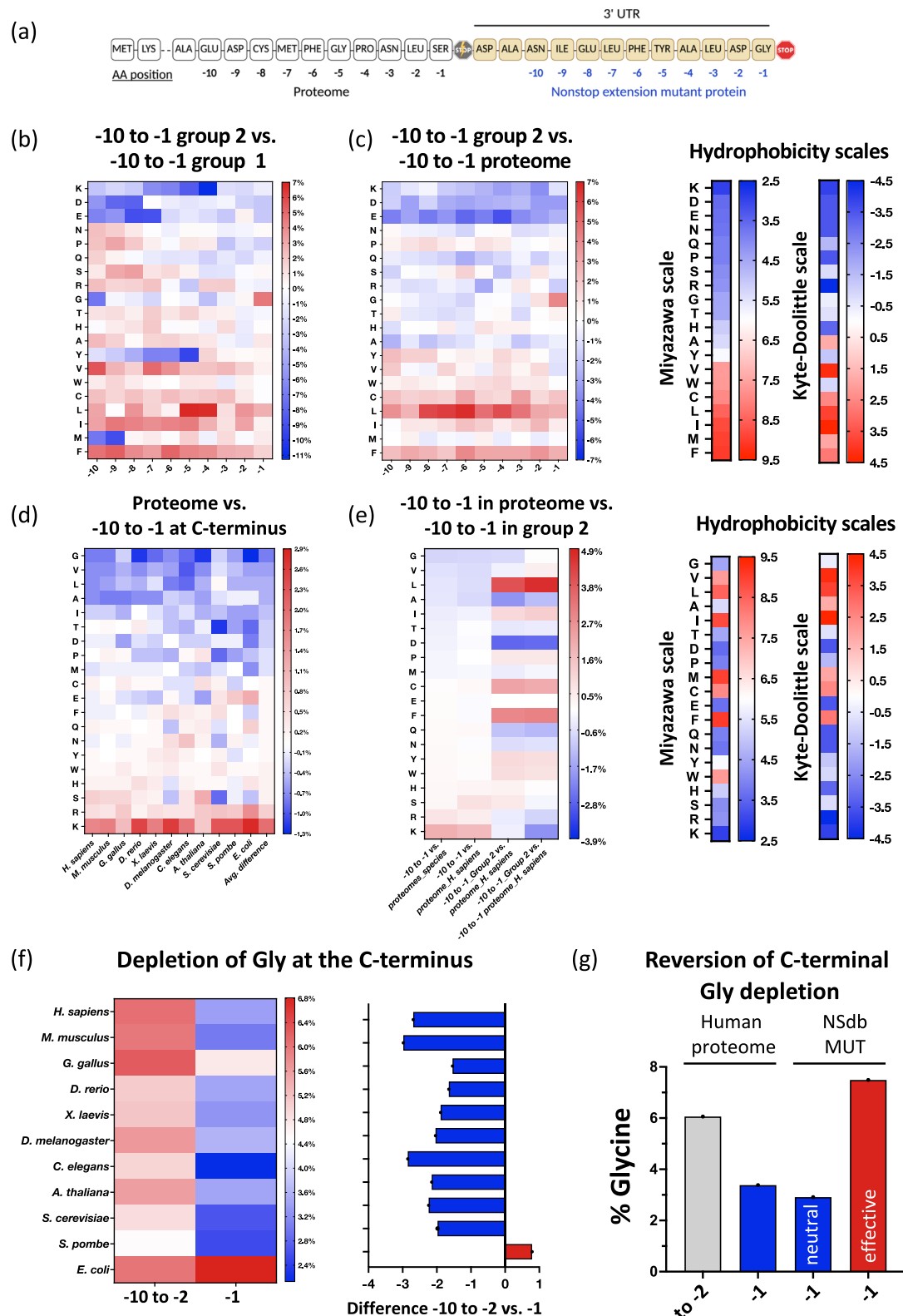

(a)

(b) **-10 to -1 group 2 vs. -10 to -1 group 1**

(c) **-10 to -1 group 2 vs. -10 to -1 proteome**

**Hydrophobicity scales**

(d) **Proteome vs. -10 to -1 at C-terminus**

(e) **-10 to -1 in proteome vs. -10 to -1 in group 2**

**Hydrophobicity scales**

(f) **Depletion of Gly at the C-terminus**

(g) **Reversion of C-terminal Gly depletion**

occurrence can have clinical impacts as observed for *SMAD4*[16], where a loss of expression could result in resistance to therapy[45]. However, beyond *SMAD4*, no nonstop mutations in cancer have been functionally characterized and their general impact remains unknown. While there are tools to predict the effect of variants of unknown significance in silico, they are generally intended for point mutations in coding regions of the genome and their accuracy for nonstop extension mutations appears to be limited, potentially because they generally

only take the mutation itself into account but not the added amino acid sequence of the extension. Effective and neutral extensions categorized by functional screening were similarly distributed between oncogenic and benign classifications by prediction tools highlighting the urgent need for experimental characterization of cancer-derived nonstop mutations to ascertain their cellular impact.

Using a high-throughput flow cytometry and next generation sequencing based screening platform, we uncover the effect of all 2335

**Fig. 6 | The amino acid composition at the C-terminus differs from the proteome across species and kingdoms and is disrupted by cancer-associated nonstop extension mutations. a** The nonstop mutation extensions led to the creation of a new C-terminal region (blue, NSdb_MUT) in comparison to the unaltered proteome (black, also NSdb_WT). The figure was created in BioRender. Ghosh, A. (2024) BioRender.com/c66h790. **b–c** Group 2 extensions showed an altered sequence composition at the C-terminus with enrichment (red) of hydrophobic residues and the glycine end degron, both, in comparison to neutral extensions of group 1 (**b**) and the human proteome (**c**). Miyazawa and Kyte-Doolittle scales of hydrophobicity are depicted for comparison. **d** The C-termini of proteomes of eleven species across kingdoms were evolutionarily conserved with an enrichment of residues like lysine (K) and arginine (R) (red) and a depletion of

residues like cysteine (C), leucine (L), and isoleucine (I) (blue). Miyazawa and Kyte-Doolittle scales of hydrophobicity are depicted for comparison. **e** The C-termini of proteomes were depleted for hydrophobic residues and enriched in hydrophilic amino acids. Contrastingly, an inverse pattern was observed for the C-termini of effective nonstop extensions, with enrichment of hydrophobic amino acids like cysteine or leucine and depletion of hydrophilic residues like lysine or arginine. **f–g** A depletion of terminal glycine (G at −1 position) was seen in all eukaryotic proteomes (**f**). In contrast, the creation of new C-termini by effective nonstop mutation extensions showed an enrichment of terminal glycine, thereby reversing the evolutionarily selected pattern at the C-termini of proteomes (**g**). In panels **b–f**, the scale depicts the normalized differences in frequency.

somatic nonstop extension mutations in cancer, cataloged in the NonStopDB. Our screen effectively shows that the majority of C-terminal nonstop extensions have a negative impact on protein expression, with a significant proportion of effective extensions eliciting a strong loss of protein expression. We highlight the importance of ten nonstop extension mutations in important CGC tier 1 tumor suppressor genes, which lead to a loss of protein expression as a result of nonstop mutations.

The loss of several tumor suppressor proteins by nonstop mutations, as verified for B2M, CDKN1B (p27KIP1) and MLH1 in this study, could have a relevant functional impact in cancer development and therapy[46–54]. This functional impact was further exemplified for *PTEN*. The decreased expression of PTEN and an increase in phosphorylation of its downstream factor AKT in case of the heterozygous stop-loss mutant cell lines demonstrates a functional effect. The upregulation of phospho-AKT in the mutant cell lines, which is known to induce the activation of the mTOR signaling cascade, underlines again the possible functional relevance of nonstop mutations in the *PTEN* gene. Several studies have elucidated that haploinsufficiency of *PTEN* is sufficient to promote or accelerate cancer in some tissues[37,55–60], indicating the relevance of our heterozygous mutant clones. Further, *PTEN* copy number loss can lead to Cetuximab resistance, once again highlighting the importance of characterizing all cancer variants[61].

Aberrant proteins are generally cycled through the proteasomal pathway[4,10,12,16,29,62] and C-terminal hydrophobic stretches have recently been linked to proteasomal degradation[12,16]. Thus, the observed rescue of PTEN expression by the proteasome inhibitor Bortezomib, thereby establishing a link with the ubiquitin-proteasome degradation pathway, is consistent with previous observations.

Our screen reveals C-terminal hydrophobicity and length to be determinants for the instability of proteins C-terminally extended by cancer-derived nonstop mutations. This association with hydrophobicity and length aligns with the previously demonstrated inherent higher hydrophobicity of the 3′ UTRs, due to the higher Uracil (U) content in the non-coding regions of the genome in contrast to the more GC-rich protein-coding sequences[10,12,62].

The increased hydrophobicity of C-terminal extensions in tumor suppressor genes as compared to those in oncogenes may be surprising from an evolutionary point-of-view as one would not expect the genome to be selected for a pro-tumorigenic potential. However, the screen had been designed to test nonstop extensions, which had already been found in cancer. Thus, their appearance may represent the selective pressure during tumorigenesis rather than evolution.

However, it is conceivable that other mechanisms could be involved in the regulation of aberrant C-terminally extended proteins, ranging from protein aggregation, protein stability disruption, protein folding, ribosome stalling, altered mRNA structure in the 3′UTR and differential microRNA binding site availability and that these mechanisms may also be responsible for regulating a proportion of the extension sequences analyzed in this screen.

Lastly, in line with previous observations, we find differences in the amino acid distribution between the proteomes and the C-termini

of proteins in eleven species across all kingdoms of life. Evolutionary selection processes constrain the amino acid composition at the C-termini of proteins, depleting destabilizing hydrophobic amino acids or C-terminal degrons like glycine[29,43], and favoring positively charged, hydrophilic amino acids[62–64]. Importantly, we uncover that cancer-derived nonstop extension mutations disrupt these highly conserved patterns by driving translation into the otherwise untranslated 3′ region and thereby creating new C-terminal sequences.

In summary, we characterize the impact of all C-terminal extensions arising from somatic nonstop or stop-loss mutations in cancer cataloged in the NonStopDB, highlighting the importance of these non-canonical mutations in cancer genetics, and proving that the majority of these can have a suppressive impact on protein expression. As more sequencing data becomes available, additional investigations may be needed to fully comprehend the underlying mechanisms behind nonstop mutation-induced phenotypes. Nonetheless, our screen can provide a foundational basis for further exploration of cancer-associated stop-loss mutations and could facilitate future studies arising from the continually expanding repositories of cancer sequences.

## Methods

### Cell culture, reagents, and treatments
The HEK293T cell line was obtained from the American Type Culture Collection (ATCC). The cells were maintained in DMEM (Sigma Aldrich) supplemented with 10% FBS (Thermo Fisher Scientific) and 2 mM L-glutamine (Sigma Aldrich). Cell lines used in this study were confirmed to be negative for mycoplasma contamination and routinely tested using the MycoGuard Mycoplasma PCR Detection Kit (Biocat). Where indicated, cells were treated with 0.1 μM Bortezomib (Calbiochem; 504314) or 2.5 μg/mL Puromycin (VWR).

### Plasmids
**Nonstop extension screen.** The dual fluorescent reporter for high-throughput analysis of the nonstop extensions was created using a lentiviral plasmid vector with an EF1α promoter, Ampicillin resistance, an IRES (internal ribosomal entry sequence) and a T2A-Puromycin selection cassette. The eGFP CDS and mCherry CDS were cloned in with Gibson assembly using a 2x Gibson assembly master mix (NEB). A cloning site comprising two Esp3I sites and a NotI site was introduced between the eGFP CDS and the IRES sequence. Additionally, a 24 nt primer binding site (PBS) was also included before the start of the IRES to facilitate PCR amplification for NGS. At the end, the dual fluorescent reporter had the following architecture: EF1α-eGFP-Esp3I-Esp3I-NotI-PBS-IRES-mCherry-T2A-Puromycin.

**TSG Validation.** The extensions used for validation of the TSGs and the *SMAD4* positive controls were also cloned into the dual fluorescent reporter plasmid using Gibson assembly. The extensions of the TSGs to be validated were synthesized as individual gene fragments from Twist Biosciences. The gene fragments had the same 24 nt Gibson assembly overhangs complementary to the 3′ end of eGFP and the 24 nt PBS, along

with a filler and adapter sequences to meet the synthesis minimum of 300 nt. Prior to cloning into the dual fluorescent reporter plasmid, only the extension containing part of the gene fragments were amplified using FW_Library and RV Validation_TSGs primers (Suppl. Data 3).

**PTEN dual fluorescent reporters with each of the 20 amino acids and interrupted "IFFY" motifs.** The dual fluorescent reporter containing the PTEN * > C or *> L extensions were used to generate the plasmids encoding each of the 20 possible amino acids by site-directed mutagenesis (SDM) using the Q5® Site-Directed Mutagenesis Kit (NEB). Similarly, the *PTEN* * > C or *>L extensions were used to generate the plasmids bearing the I > E change in the *PTEN* extension by SDM. The primers used for the SDM are listed in Suppl. Data 3.

**PTEN overexpression plasmids.** The *PTEN* CDS with the respective C-terminal extensions were cloned into the same lentiviral backbone containing an Ampicillin resistance cassette as used for the library. The vector also included a Kozak sequence, a FLAG tag, an HA tag, and a Glycine-Serine (GS) linker. The vector was assembled using a two fragment Gibson assembly with one fragment coding for the Kozak sequence, FLAG tag, HA tag and GS linker and the other encoding the *PTEN* CDS. The *PTEN* CDS was amplified using a two-step PCR strategy wherein the first PCR, using the FW_PTEN_CDS and RV_PT_Puro primers, amplified the *PTEN* CDS and the second PCR using the same forward primer and the RV_GA_PTEN_CDS primer introduced a Gibson assembly arm. The final *PTEN* overexpression plasmid had the following architecture: EF1α-Kozak-FLAG tag-HA tag-GS linker-*PTEN* CDS-Extension-SV40 promoter-Puromycin.

**Group 1 overexpression plasmids.** The CDS for *GATA1* was amplified from the TFORF2242 plasmid using FW_GATA1_CDS and RV_GA-TA1_CDS. The CDS was then cloned into the same lentiviral backbone containing an Ampicillin resistance cassette as used for the library. The vector also included a Kozak sequence, a FLAG tag, an HA tag, and a Glycine-Serine (GS) linker. The vector was assembled using a three fragment Gibson assembly with one fragment coding for the Kozak sequence, FLAG tag, HA tag and GS linker, the second encoding the *GATA1* CDS and the third comprising the plasmid backbone. The fragment containing the Kozak sequence, FLAG and HA tags and GS linker was amplified from the FW-kz-FLAG-HA-GS linker and RV-kz-FLAG-HA-GS linker using the FW_tag and RV_tag primers. The nonstop mutations were introduced using by SDM using the Q5® Site-Directed Mutagenesis Kit (NEB).

The pLenti-6 plasmids with the wildtype and nonstop mutation CDS and extension of *ACO2* and *PRKCH* were cloned as previously described for *SMAD4*[16]. These plasmids were used to amplify the wildtype or mutant CDS along with the extension using the FW_A-CO2_CDS and RV_ACO2_CDS and FW_PRKCH_CDS and RV_PRKCH_CDS primers for *ACO2* and *PRKCH*, respectively. The amplified CDS and extension for ACO2 and PRKCH were then cloned into the lentiviral vector used for the library via a three fragment Gibson assembly with one fragment coding for the Kozak sequence, FLAG tag, HA tag and GS linker, the second encoding the respective CDS with the extension and the third comprising the plasmid backbone.

The TFORF2242[65] was a gift from Feng Zhang (Addgene plasmid #141986).

**Group 2 overexpression plasmids.** The CDSs for *B2M*, *MLH1* and *CDKN1B* were amplified from the plasmids pHAGE-B2M[66], pCEP9-MLH1[67] and p27-IRES-mCherry, respectively[68]. The C-terminal extension sequences occurring as a result of the nonstop mutations were synthesized as gene fragments from IDT (Suppl. Data 3). The CDS and gene fragments with the C-terminal extension were cloned into the lentiviral backbone using a four fragment Gibson assembly similar to the other overexpression plasmids. The final plasmids contained a Kozak sequence-FLAG tag and GS linker followed by the respective HA-tagged wildtype CDS with the C-terminal extension sequence. SDM using the Q5® Site-Directed Mutagenesis Kit (NEB) was used to create the nonstop mutation plasmids from the wildtype plasmids.

The group 1 and group 2 overexpression plasmids were sequence-verified using the FW_overepxression_seq along with CDS specific primers as listed in Suppl. Data 3.

The pHAGE-B2M plasmid was a gift from Gordon Mills & Kenneth Scott[66] (Addgene plasmid #116715), the pCEP9-MLH1 plasmid was a gift from Bert Vogelstein[67] (Addgene plasmid #16458), and the p27-IRES-mCherry plasmid was a gift from Ellen Rothenberg[68] (Addgene plasmid #80143).

All plasmids were transformed into NEB Stable competent *E. coli* (Thermo Fisher Scientific).

A complete list of primers and sequences used in the study can be found in Suppl. Data 3.

## Production and next generation sequencing of the stop-loss extension reporter library

**Production.** For the nonstop extension library, a pool of 4735 oligonucleotides (oligo pool), each 300 nt long, was synthesized by Twist Biosciences. The sequences of the nonstop extensions were obtained from the NonStopDB and each sequence was designed to end with a stop codon. To ensure a size balanced library with all oligos having a length of 300 nt, an inert filler sequence was added to the nonstop extension sequences shorter than 252 nt. Each oligonucleotide in the pool contained a 24 nt constant sequence on each side for Gibson assembly flanking the variable nonstop extension sequences. The left constant sequence was the last 24 nt of the eGFP CDS excluding the stop codon (5'-CTCGGCATGGACGAGCTGTACAAG-3'). The right constant sequence comprised the PBS included during plasmid construction (5'-GCGGCCGCACAGAGTGCACGTCGT-3').

The oligo pool was resuspended to a concentration of 10 ng/μL and PCR amplified using the NEBNext Ultra II Q5® Master Mix (NEB) and the FW_library and RV_library primers. The amplified oligo pool was then gel extracted using the GeneJet gel extraction kit (Thermo Fisher Scientific) and further purified and concentrated using AmpureXP beads (Beckman Colter). The concentration of the purified oligo pool was measured on a Qubit 3.0 (Thermo Fisher Scientific) using the Qubit dsDNA BR (Broad-Range) Assay Kit (Thermo Fisher Scientific). The purified oligo pool was then cloned into the Esp3I digested dual fluorescent reporter plasmid by Gibson assembly. The Esp3I digestion of the dual fluorescent reporter plasmid, cleaves out the eGFP CDS stop codon, allowing the stop-loss extensions of the oligo pool to be cloned directly to the 3' end of the eGFP CDS. The Gibson assembly reaction containing the cloned nonstop extension plasmid library was then transformed into ElectroMAX™ DH10B Cells (Thermo Fisher Scientific) using the Micropulser Electroporator (BioRad). The transformed bacteria at a dilution of $10^{-2}$ plated on twenty-five 15 cm LB-Ampicillin plates to ensure a coverage of 2000x. 24 h after plating, the colonies were scraped from each of the 25 plates using 7 mL LB medium per plate and used to perform a Maxiprep with the PureLink™ HiPure Plasmid Maxiprep Kit (Thermo Fisher Scientific) to obtain the plasmid library of nonstop extensions.

The maxiprep containing the plasmid library was then analyzed using NGS. For this, a two-step PCR amplification using the NEBNext Ultra II Q5® Master Mix (15 cycles each) (NEB) was performed. The first PCR was performed using the FW_library and RV_library primers that target the left constant region of the oligo pool and the 24 nt PBS included during cloning. The second PCR reaction was performed using forward degenerate primers (FW_NGS_deg6 - FW_NGS_deg9) that included the TruSeq Universal adapter P5 and a reverse Illumina sequencing primer that included the Illumina index sequence (RV_NGS_index8- RV_NGS_index10), with 10 μL of PCR product from the first PCR reaction as template. The PCR amplicon from the second

PCR reaction was then gel extracted using the GeneJet gel extraction kit (Thermo Fisher Scientific) and further purified and concentrated using AmpureXP beads (Beckman Colter). The concentration of the purified amplicon was measured on a Qubit 3.0 (Thermo Fisher Scientific) using the Qubit dsDNA HS (High sensitivity) Assay Kit (Thermo Fisher Scientific). 10 pmol of the second PCR amplicon was sequenced in the forward direction with a 10% PhiX control v3 (Illumina) spike-in on an Illumina MiSeq using the MiSeq Reagent Kit v3 (150-cycle) (Illumina).

**Inert sequence validation.** Prior to inclusion of the inert sequence included for the library synthesis, the sequence was analyzed to ensure that the sequence by itself does not trigger a loss of eGFP. For this, the *SMAD4* WT and *SMAD4* * > C extension sequences were synthesized with and without the inert sequence in the form of gene blocks from Integrated DNA Technologies (IDT). The gene blocks were designed to have the same Gibson assembly arms as in the library. The lyophilized gene blocks were resuspended and used for cloning using Gibson assembly into the dual fluorescent reporter plasmid. These plasmids were then analyzed by transient transfection and flow cytometry to determine the effect on eGFP expression.

## Virus production, transient transfections

For virus production, HEK293T cells were seeded at a density of $2 \times 10^6$ cells per plate in 10 cm dishes and transfected with a total of 20 μg plasmid per 10 cm dish. The cells were transfected in Opti-MEM (Gibco) with the second-generation lentiviral packaging plasmids, psPAX and PMD2.G and the lentiviral plasmid library at a ratio of 2:1:4, using PEI (Sigma Aldrich). Viral supernatants were collected 24 h and 48 h after the transfection, pooled and filtered through 0.45 μm filters (Merck). The virus supernatants were flash frozen in liquid nitrogen and stored at −80 °C. To calculate the MOI needed for a 30% transduction efficiency, the virus supernatants were assessed via transduction into HEK293T cells. For this, HEK293T cells were seeded at $0.5 \times 10^6$ cells per well in 6- well plates. 24 h after seeding, the cells were transduced with varying volumes of virus supernatant ranging from 50 to 1000 μL, in the presence of 8 μg/mL Polybrene (Sigma Aldrich). 24 h after transduction, the cells were selected using 2.5 μg/mL Puromycin for 48 h. The transduction efficiency was then determined by flow cytometry.

For transient transfections, HEK293T cells were transfected using PEI (Sigma Aldrich). 24 h prior to transfection, the cells were seeded in either 12- or 24-well plates at a density of $0.5 \times 10^6$ or $0.05 \times 10^6$ cells per well, respectively. Plasmids were diluted in Opti-MEM (Gibco), mixed with PEI at a ratio of 1.5 μl PEI per 500 ng plasmid DNA and incubated at room temperature for 15 min. Fresh Opti-MEM was added to the wells along with the DNA-PEI complexes. The transfection complex containing Opti-MEM was changed to complete medium 4–6 h after transfection. Two days after transfection, the cells were selected with 2.5 μg/mL Puromycin for 48 h. Flow cytometry analyzes of transfected cells were typically performed 96 h after transfection.

## High throughput screen for nonstop mutations in HEK293T cells

**Library transduction.** The nonstop extension library was used to generate cell libraries in HEK293T using lentiviral transduction such that each cell was integrated with at most one virus by a multiplicity of infection (MOI) of 0.3. For the screen, we aimed for a 2000x coverage. 24 h prior to transduction, HEK293T cells were seeded into four 15 cm dishes at $7.5 \times 10^6$ cells / dish. The cells were then transduced with filtered supernatants of the viral library for 24 h along with 8 μg/ml polybrene (Sigma Aldrich). 48 h after transduction, the cells from each 15 cm dish were trypsinized using 0.05% trypsin-EDTA (Life Technologies) and transferred to a T-225 flask containing medium with 2.5 μg/mL Puromycin. 24 h after transfer to the T-225 flask, fresh medium with 2.5 μg/mL Puromycin was added. 96 h after transduction, the cells were

processed for fluorescence activated cell sorting (FACS) with the BD FACS Aria (BD Biosciences). The cells were sorted into three populations: mCherry high & eGFP low (eGFP low); mCherry high & eGFP intermediate (eGFP intermediate); mCherry high & eGFP high (eGFP high) at approximately $8 - 10 \times 10^6$ cells per population. Genomic DNA was separately isolated from each of the populations using the DNeasy Blood & Tissue Kit (Qiagen). The nonstop extensions in each population were then PCR amplified using the two-step PCR strategy described for the plasmid library sequencing using the NEBNext Ultra II Q5® Master Mix (15 cycles each) with each of the four populations possessing a unique index. 10 pmol of the gel-extracted and Ampure XP-purified second PCR amplicons from each population was sequenced with a 10% PhiX control v3 spike-in on an Illumina MiSeq using the MiSeq Reagent Kit v3 (150-cycle) (Illumina). Four independent biological replicates of the screen were performed.

**NGS analysis.** The quality of the sequencing and the analysis of the fastq files for each of the populations was performed using Galaxy[69]. The reads in each population were processed using Cutadapt[70] and Trimmomatic[71] and the read count for each extension in the library was performed using MAGeCK count[72]. For each population, the read counts were normalized to the total read counts in the sequencing run to obtain a normalized read count per million (RPM).

For each extension, we performed the analysis as described in the flow chart in Fig. S1d. In brief, we analyzed the number of normalized reads in the eGFP low and eGFP intermediate populations as a fraction of the total number of reads in all three populations. Further, to look at the fold differences between the various nonstop extensions in our screen, we used a ratio of the normalized read counts in either the eGFP low or intermediate populations to the normalized read count in the eGFP high population for each oligo in each biological replicate to obtain an enrichment score. We then used the median of the enrichment scores of the four biological replicates to divide our library into four groups: group 1 = neutral: extensions without any effect on eGFP, where the median enrichment was ≤ 2 and the sum of all reads was ≥ 100; group 2 = highly effective: extensions that destabilize eGFP, where the median enrichment was ≥ 3 in at least three of the four replicates and the sum of all reads was ≥ 100; group 3 = moderately effective: extensions that showed a moderate loss of eGFP, where the median enrichment was between 2 and 3 or the replicates deviated from the other two groups 1 and 2 and the sum of all reads was ≥ 100 and group 4 = extensions with very low expression where the sum of all reads was < 100.

## Flow cytometry analysis

HEK293T cells were washed with 1x PBS (Sigma Aldrich), trypsinized with 0.05 % trypsin-EDTA and collected in 500 μL to 1 mL of complete medium. The cells were centrifuged at 100 g for 5 min and then resuspended in 500 μL to 1 mL of 1x PBS. The cells were passed through a 100 μM mesh cell strainer (Sysmex) immediately prior to flow cytometry. Flow cytometry was performed on a BD LSR Fortessa (BD Biosciences). The live cells and singlets were gated from the total cell population, followed by the quadrant gating of the eGFP-mCherry positive cells. The gating strategy is described in Fig. S2d. For analyzing the eGFP/mCherry ratio, the channel values for each cell expressing mCherry in the gated live cell population were exported. Gating and export of data for downstream analysis was done using the FlowJo v10.8.1 software (BD Biosciences).

## CRISPR genome editing

Two guide RNAs were designed in the proximity of the stop codon of the *PTEN* gene to create the stop-to-leucine (*> L) (TGA > TTA) mutation. The guide RNAs were synthesized as 43 nt Alt-R™ A.s. Cas12a crRNAs (CRISPR RNAs) by IDT. For editing, the guide RNAs were complexed with Alt-R™ A.s.Cas12a (Cpf1) Ultra (IDT) as per IDTs

guidelines to form a ribonucleoprotein particle (RNP). A 200 nt Alt-R™ HDR Donor Oligo encoding the *> L edit, synthesized by IDT, was used as the homology directed donor repair template. The RNPs were delivered into HEK293T cells by nucleofection according to previously described methods[73–75] using the SF Cell Line 4D-Nucleofector™ X Kit (Lonza) and the Amaxa 4D-Nucleofector device (Lonza).

In brief, HEK293T cells grown to 70% confluency were washed with 1x PBS, trypsinized and counted. The total number of cells needed were transferred to a sterile 15 mL tube and centrifuged at 100 g for 5 min at room temperature. The supernatant was removed, and the pellet was washed with 5 mL of 1x PBS followed by centrifugation at 100 g for 5 min at room temperature. The cells were resuspended in 20 µL of supplemented nucleofector solution SF (Lonza) per $3.5 \times 10^5$ cells. For each reaction, 20 µL of resuspended cells was mixed with 5 µL of the pre-formed RNP containing one of the two guide RNAs and 78 µM Alt-R® Cpf1 Electroporation Enhancer (IDT). Additionally, 120 pmol of the HDR donor oligo was added. The final mixture was pipetted up and down two times and carefully transferred to separate wells of a 16-well Nucleocuvette™ strip. The Nucleocuvette™ strip was placed into the 4D-Nucleofector™ X Unit and the cells were nucleofected using the CM130 program. The Nucleocuvette™ strip was removed from the device and 80 µL of pre-warmed culture media was added to each of the nucleofected wells in the Nucleocuvette™ strip. The cells were resuspended by gentle pipetting and transferred to a prepared 12-well plate containing 1 mL culture media and the 30 µM Alt-R™ HDR Enhancer V2 (IDT). The cells were incubated in a humidified 5% $CO_2$ incubator at 37 °C.

48 h post nucleofection, the cells were trypsinized and part of the cells were harvested for genomic DNA isolation using the DNeasy Blood & Tissue Kit (Qiagen). The genomic DNA was amplified around the editing locus using FW_PTEN_seq and RV_PTEN_seq and sequenced to assess the presence of the desired mutation. The remaining cells were diluted and used for single cell isolation in 96-well plates. The successfully edited heterozygous PTEN mutant cell lines (Clone #1 and Clone #2) were established and the mutation was confirmed again with Sanger sequencing. The chromatogram figure depicting the mutation in Fig. S3k was created using Benchling (www.benchling.com, accessed on 26 Feb 2024).

### Western Blotting and image quantification

Cells were cultured and transfected as described above. Cells were washed with cold PBS and lysed in RIPA buffer (150 mM NaCl, 50 mM Tris pH 7.4, 0.1% SDS, 0.5% Sodium deoxycholate, 1% Triton X-100) supplemented with a 1x protease inhibitor cocktail (Promega) and 1x PhosSTOP (Roche) for 60 min on ice. Lysates were then cleared by centrifugation at 16,000 g and 4 °C for 30 min. Protein concentrations were determined using a Bradford assay (Serva Electrophoresis GmbH) and the required volumes of samples were prepared in RIPA buffer and 6x SDS loading buffer (0.5 M Tris (pH 6.8), 50% glycerol, 10% DTT (Dithiothreitol), 0.001% Bromophenol blue and 4% SDS) and heated to 95 °C for 10 min. Samples were then run on a 10% SDS-PAGE gel and transferred to an activated 0.45 µM PVDF membrane for 90 min at 80 V. Membranes were blocked with 5% milk solution prepared in 1% TBST (1x TBS and 0.1% Tween-20) for 1 h at room temperature. Membranes were then cut and incubated overnight at 4 °C, with the appropriate primary antibody at a 1:1000 dilution prepared in either in 5% milk with TBST or 5% BSA (Bovine Serum Albumin) with TBST. Secondary antibodies were added at a 1:8000 dilution and incubated for 1 h at room temperature. The blots were visualized using Amersham ECL Prime Western Blotting Detection Reagent and the ChemoCam Imager (INTAS) and the bands were quantified using FusionCapt Advance software v17.04 (Vilber). Primary antibody against the HA-tag (BioLegend, Antibody #901501) was prepared with 5% milk in 1x TBST and then probed with an anti-mouse antibody (Dianova, 115-035-003). The primary antibodies against *PTEN* (Cell Signaling Technology, Antibody #9188), pAKT-Thr308 (Cell Signaling Technology, Antibody #9275), panAkt (Cell Signaling Technology, Antibody #4691) and Vinculin (Cell Signaling Technology, Antibody #13901) were prepared in 5% BSA in 1x TBST and then further probed with an anti-rabbit antibody (Dianova, 111-035-003).

### Statistics and reproducibility

Raw NGS read counts for the high-throughput screen are available in Suppl. Data 1. All experimental values for the high-throughput screen are presented as median with 95% confidence intervals (95% CI). For the screening data, statistical analyses were done using the non-parametric Mann-Whitney U test. The experimental values for validation experiments using individual eGFP fusion reporters are presented as mean ± SEM and the statistical analyses were done using analysis of variance (ANOVA). GraphPad Prism version 10.2.0 for MAC (GraphPad Software, Boston, Massachusetts USA, www.graphpad.com) was used for creating all graphs and for performing the statistical tests. 'n' in the figure captions denotes the number of biologically independent replicate experiments. All experiments were carried out in at least three independent biological replicates. Schematics in Figs. 1a, b, Fig. 6a, Fig. S1b, d, and Fig. S3i–j were created using Biorender.com.

### Reporting summary

Further information on research design is available in the Nature Portfolio Reporting Summary linked to this article.

## Data availability

The nonstop extension sequences used in this study were obtained from and is provided in the NonStopDB database accessible at https://NonStopDB.dkfz.de. The raw NGS data has been deposited into the NCBI Sequence Read Archive (SRA) under Bioproject accession PRJNA1099128. The raw NGS read counts for the high-throughput screen is available in Suppl. Data 1. All other data generated or analyzed as part of this study are included within this published article and its supplementary files. Source data are provided with this paper.

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

## Acknowledgements

This research was funded by the German Research Foundation (DFG Di 1421/9-2). We are grateful to Dr. Dietmar Pfeifer and Andreas Janes for the generous use of their MiSeq machine. We also thank the Lighthouse Core Facility at the Center for Translational Cell Research (ZTZ) in Freiburg for support with the flow cytometry experiments and cell sorting, and the Freiburg Galaxy team with Dr. Björn Grüning and Prof. Dr. Rolf Backofen for providing this resource.

## Author contributions

S.Di. conceived the study, and S.Di. and A.G. designed the study. S.Di. and A.G. designed the experiments, and A.G., M.R., K.L., and A.W. performed the experiments. S.Dh. and U.K.D. cloned the plenti6 plasmids for *ACO2*, *GATA1* and *PRKCH*. J.P. and C.T. contributed to project discussions. S.Di. and A.G. analyzed the data and wrote the manuscript, and A.G. prepared the figures. All authors have read the manuscript and approved the final version.

## Funding

## Competing interests

S.Di. is a co-owner of siTOOLs Biotech, Martinsried, Germany, unrelated to this work. The other authors declare no competing interests.
