## [Peer Review File · Nature Communications]

REVIEWER COMMENTS

Reviewer #1 (Remarks to the Author): Expert in functional genomics, gene expression regulation, evolution, and molecular genetics

Using massively parallel reporter assays, Ghosh et al systematically analyzed the impact of all >2000 nonstop somatic mutations they previously cataloged. They found that most nonstop mutations resulted in the degradation of the extended protein due to the presence of hydrophobic amino acids encoded by the 3' UTR sequence. Nonstop extension in tumor suppressor genes appears to be more potent in inducing protein degradation, compared to oncogenes and other genes in the genome. Analysis of 11 species also revealed amino acid biases in protein C-termini.

This study is a significant advance in our understanding of tumor mutations. While individual non-stop mutations have been studied - including their previous seminal study on SMAD4 - this is the first comprehensive study testing the impact of all nonstop mutations in cancer. The study is well designed and the manuscript is well written. I only have a few minor comments.

Two minor comments about the title: "Effective cancer nonstop extension mutations increase C-terminal hydrophobicity and disrupt evolutionarily conserved amino acid patterns". The current title focuses on hydrophobicity but does not mention protein degradation, which is in my opinion a more important observation of the paper, i.e., most somatic non-stop mutations cause protein degradation. The current title also implies that the statement is true for individual genes, i.e., the hydrophobicity of the new C-terminus (extension) created by a nonstop mutation is higher than that of the wild type protein of the same gene. This analysis is missing in the manuscript and would be important if the current title is to be used.

The observation that nonstop extensions in tumor suppressor genes are more likely to cause protein degradation than that of oncogene is intriguing. While this difference makes sense from the perspective of tumorigenesis, the human genome is not evolved for tumor to happen. If anything, the opposite trend should be expected, i.e., the extension downstream of tumor suppressors should have evolved to encode less hydrophobic sequences to avoid degradation and subsequent tumor formation in case of nonstop mutation. A discussion on potential driving forces behind the observed difference will be helpful. Please also show whether the extension of tumor suppressors are more hydrophobic than that of oncogenes.

Reviewer #2 (Remarks to the Author): Expert in functional cancer genomics and high-throughput mutation screening

This manuscript describes a high-throughput analysis of stop-loss mutations derived from the human cancer database (i.e. COSMIC) using HEK293T-EGFP reporter cell lines. The authors first generated HEK293T cell library expressing EGFP fused with intended C-terminal extensions library, containing 2335

nonstop mutations with wildtype controls, thereby enabling them to evaluate the impact of nonstop mutation and C-terminal extension on protein expression. They quantitatively assessed the impact of nonstop mutation by calculating median enrichment (M.E.) scores, which represent the fraction of cells containing each mutation observed in EGFP-reduced populations to EGFP-high populations. They individually validated the impact of several nonstop mutations in the EGFP reporter system and exogenous expression system of PTEN. In addition, they generated clonal cells harboring PTEN nonstop mutations and showed decreased protein expression and increased level of AKT phosphorylation. Next, they analyzed the factors affecting suppressive effects of translations by nonstop mutations, and revealed hydrophobicity and length of C-terminal extension is key factors on their effect on protein expression.

A significant portion of cancer variants remains classified as variants of uncertain significant (VUSs). Previous reports demonstrated most noncoding translations eventually lead to C-terminal extension and subsequent protein degradation (PMID: 37046090). The authors of this study have presented similar findings, showing that nonstop mutations in cancer-related genes result in the loss of gene function. In contrast to previous report (PMID: 37046090) which utilized a similar reporter system and focused on nonstop mutations in noncoding sequences such as introns and UTRs, the authors of this study placed more emphasis on nonstop mutations found in cancer databases. Overall, I found that this manuscript is well-designed and provides significant insight on nonstop mutations in human cancer. However, there are several technical issues that should be addressed and suggestion.

1. Generally, identifying a cancer driver mutation is determined by its frequency in cancer samples (PMID: 29625053). Therefore, revealing the frequency of nonstop mutations in well-known tumor suppressor genes supports the importance of the study. However, there seems to be no mention of the frequency of nonstop mutations in this manuscript or the authors' previous paper (PMID: 32719554) (I may have missed it), and the authors did not mention the frequency of nonstop mutations in cancer, at least in well-known cancer related genes. I think it would be helpful to investigate tumor suppressor genes (TSGs) or cancer census genes (PMID: 30293088) to strengthen the significance of this paper. Furthermore, is there variation in the frequency of these nonstop mutations across different types of cancer?

2. In figure 2d, the authors demonstrate that the distribution of median enrichment score between group 1 and group 2 differed significantly. However, more quantitative analysis on the validity of M.E. score would strengthen the study. For example, could the authors perform receiver-operating characteristics (ROC) analysis of Median enrichment scores targeting canonical positive control (i.e. known degron or validated positive controls) versus negative control? In addition, conducting ROC analysis using M.E scores and other computationally predicted scores such as CADD or FATHMM and comparing their performance would be informative.

3. The evidence presented in the current manuscript is not sufficient to show whether the M.E. score obtained from the reporter system directly reflects the extent of protein degradation within cells. The authors evaluated protein expression only for three mutations observed in the PTEN gene and for C-terminal extensions in Figure 4a,b. However, this analysis includes only a few mutations that emerged as hits and does not include mutations with various ranges of M.E. scores as controls. I suggest that the authors include several additional controls, not just one PTEN gene, to confirm the correlation between

the M.E. score obtained from high throughput screening and the extent of protein expression degradation.

4. The authors suggested several features that determine nonstop-mediated loss of expression, such as length of the neutral extension, predicted hydrophobicity, composition of hydrophobic amino acids, existence of glycine at the last position, etc. Could the authors perform feature importance analysis (i.e. SHAP value) on various features suggested from the high-throughput analysis? This analysis could serve as important evidence for predicting the pathogenicity of nonstop mutations in the future.

Response Letter

First, we would like to thank the editor and the reviewers for their positive, thoughtful and encouraging evaluation of our study. We have now carefully revised the manuscript according to the suggestions and have addressed all points as detailed below. Changes in the manuscript are highlighted in blue.

Reviewer #1

Using massively parallel reporter assays, Ghosh et al systematically analyzed the impact of all >2000 nonstop somatic mutations they previously cataloged. They found that most nonstop mutations resulted in the degradation of the extended protein due to the presence of hydrophobic amino acids encoded by the 3' UTR sequence. Nonstop extension in tumor suppressor genes appears to be more potent in inducing protein degradation, compared to oncogenes and other genes in the genome. Analysis of 11 species also revealed amino acid biases in protein C-termini.

This study is a significant advance in our understanding of tumor mutations. While individual non-stop mutations have been studied - including their previous seminal study on SMAD4 - this is the first comprehensive study testing the impact of all nonstop mutations in cancer. The study is well designed and the manuscript is well written. I only have a few minor comments.

Two minor comments about the title: "Effective cancer nonstop extension mutations increase C-terminal hydrophobicity and disrupt evolutionarily conserved amino acid patterns". The current title focuses on hydrophobicity but does not mention protein degradation, which is in my opinion a more important observation of the paper, i.e., most somatic non-stop mutations cause protein degradation.

We intended to express the protein loss with the word "effective" in the title, but we agree that this was not clear enough. To stay within the length restrictions given by the journal, we have replaced this word in the title with the word "suppressive" (l. 1). We are somewhat hesitant to specifically refer to protein degradation since strictly speaking we cannot conclude on the exact mechanism of protein loss from our studies, which may also affect other stages in the protein life cycle.

The current title also implies that the statement is true for individual genes, i.e., the hydrophobicity of the new C-terminus (extension) created by a nonstop mutation is higher than that of the wild type protein of the same gene. This analysis is missing in the manuscript and would be important if the current title is to be used.

Following the reviewer's suggestion, we have added systematic analyses of all tested extensions for the change in hydrophobicity.

When we compare the hydrophobicity in the last ten C-terminal amino acids between the wildtype C-termini and the extension C-termini, we do not find any significant difference for the group of neutral extensions (group 1, figure 5i,j), but a highly significant difference for the effective extensions (group 2, figure 5k,l) for both, the Miyazawa and the Kyte-Doolittle scales for hydrophobicity.

Additionally, we compare the difference in hydrophobicity (delta) between wildtype and mutant C-termini on both hydrophobicity scales and observe a difference between the neutral and the effective group (supplementary figure S4d,e).

Thus, both of these global analyses are well in line with our observations for individual genes.

The observation that nonstop extensions in tumor suppressor genes are more likely to cause protein degradation than that of oncogene is intriguing. While this difference makes sense from the perspective of tumorigenesis, the human genome is not evolved for tumor to happen. If anything, the opposite trend should be expected, i.e., the extension downstream of tumor suppressors should have evolved to encode less hydrophobic sequences to avoid degradation and subsequent tumor formation in case of nonstop mutation. A discussion on potential driving forces behind the observed difference will be helpful.

We agree with the reviewer that the genome did likely not evolve to be pro-tumorigenic. But our screen has been designed to test nonstop extensions which had already been found in cancer - and hence potentially selected for during tumorigenesis. Thus, their appearance may rather represent the selective pressure during tumorigenesis than evolution. Also, since many tumors arise after the reproductive phase as a disease associated with age, their impact on evolutionary selection may be lower than e.g. the impact of diseases occurring earlier in life. We have added a respective paragraph in the Discussion section (l. 383-388).

Please also show whether the extension of tumor suppressors are more hydrophobic than that of oncogenes.

To further analyze the effect on tumor suppressor genes, we have added an analysis directly comparing the oncogene-derived versus the tumor suppressor gene-derived extensions included in the screen using the classification according to the Cancer Gene Census. The average hydrophobicity in extensions derived from tumor suppressor genes is significantly higher than the hydrophobicity of extensions derived from oncogenes independent of the hydrophobicity scale applied (supplementary figure S4a,b).

Reviewer #2

This manuscript describes a high-throughput analysis of stop-loss mutations derived from the human cancer database (i.e. COSMIC) using HEK293T-EGFP reporter cell lines. The authors first generated HEK293T cell library expressing EGFP fused with intended C-terminal extensions library, containing 2335 nonstop mutations with wildtype controls, thereby enabling them to evaluate the impact of nonstop mutation and C-terminal extension on protein expression. They quantitatively assessed the impact of nonstop mutation by calculating median enrichment (M.E.) scores, which represent the fraction of cells containing each mutation observed in EGFP-reduced populations to EGFP-high populations. They individually validated the impact of several nonstop mutations in the EGFP reporter system and exogenous expression system of PTEN. In addition, they generated clonal cells harboring PTEN nonstop mutations and showed decreased protein expression and increased level of AKT phosphorylation. Next, they analyzed the factors affecting suppressive effects of translations by nonstop mutations, and revealed hydrophobicity and length of C-terminal extension is key factors on their effect on protein expression.

A significant portion of cancer variants remains classified as variants of uncertain significant (VUSs). Previous reports demonstrated most noncoding translations eventually lead to C-terminal extension and subsequent protein degradation (PMID: 37046090). The authors of this study have presented similar findings, showing that nonstop mutations in cancer-related genes result in the loss of gene function. In contrast to previous report (PMID: 37046090) which utilized a similar reporter system and

focused on nonstop mutations in noncoding sequences such as introns and UTRs, the authors of this study placed more emphasis on nonstop mutations found in cancer databases. Overall, I found that this manuscript is well-designed and provides significant insight on nonstop mutations in human cancer. However, there are several technical issues that should be addressed and suggestion.

1. Generally, identifying a cancer driver mutation is determined by its frequency in cancer samples (PMID: 29625053). Therefore, revealing the frequency of nonstop mutations in well-known tumor suppressor genes supports the importance of the study. However, there seems to be no mention of the frequency of nonstop mutations in this manuscript or the authors' previous paper (PMID: 32719554) (I may have missed it), and the authors did not mention the frequency of nonstop mutations in cancer, at least in well-known cancer related genes.

To include this point into the manuscript, we now mention that the frequencies of each of these mutations as well as the tumor entities in which they have been found can be obtained from the NonStopDB database (l. 58-59).

I think it would be helpful to investigate tumor suppressor genes (TSGs) or cancer census genes (PMID: 30293088) to strengthen the significance of this paper.

In addition to the analysis comparing the effectiveness of oncogene-derived versus tumor suppressor gene-derived extensions (figure 3a), we have added an analysis directly comparing the hydrophobicity of oncogene-derived versus the tumor suppressor gene-derived extensions included in the screen using the classification according to the Cancer Gene Census. The average hydrophobicity in extensions derived from tumor suppressor genes is significantly higher than the hydrophobicity of extensions derived from oncogenes independent of the hydrophobicity scale applied (supplementary figure S4a,b).

Regarding frequency of nonstop extension mutations in cancer genes (according to the Cancer Gene Census), we have previously documented that nonstop extension mutations are similarly enriched in cancer genes as missense mutations and higher than synonymous mutations (PMID 32719554, figure 1d). We have added this information to the Introduction section (l. 59-60).

Lastly, we have validated the suppressive effect for five nonstop extension mutations in three tumor suppressor genes by western blotting (figure 3f-j)

Furthermore, is there variation in the frequency of these nonstop mutations across different types of cancer?

Following the reviewer's suggestion, we have analyzed the nonstop mutation frequency in different entities. At the tumor entity level, the pan-cancer NonStopDB distinguishes 96 different tumor entities annotated according to COSMIC v89. Most of the 3412 nonstop mutations are found in tumors of the digestive system (1255) followed by the genitourinary system (736) and the respiratory system (712). For the individual tumor entity, most nonstop mutations are found in the adenocarcinoma of the large intestine (569) followed by the adenocarcinoma of the lung (243) and carcinoma of the breast (201).

At the gene level, the genes with most nonstop extension mutations are ACO2 (13) with all mutations found in head and neck squamous cell carcinoma and PRKCH (11) with all mutations found in glioma. The cancer gene with most nonstop extension mutations is SMAD4 (11) with all mutations found in

colon, pancreatic and bile duct cancer, which are known to be driven by different loss-of-function mutations in SMAD4 (PMID 32719554, figure 1h). Thus, these represent entity-specific nonstop extension mutations. In turn, the nonstop mutations in the cancer gene CDKN2A (9) are found in different entities including lung squamous cell carcinoma (2), lung adenocarcinoma (2), lung small cell carcinoma (1), head and neck squamous cell carcinoma (1), esophagus squamous cell carcinoma (1), pancreas ductal adenocarcinoma (1) and malignant melanoma (1).

We have added this information to the Results section (l. 72-87).

2. In figure 2d, the authors demonstrate that the distribution of median enrichment score between group 1 and group 2 differed significantly. However, more quantitative analysis on the validity of M.E. score would strengthen the study. For example, could the authors perform receiver-operating characteristics (ROC) analysis of Median enrichment scores targeting canonical positive control (i.e. known degron or validated positive controls) versus negative control?

To further corroborate the validity of the screen, we have performed the suggested ROC analysis comparing positive controls (degrons, SMAD4 extensions) versus negative controls (stop codons, wildtype extension sequences with the intact stop codon). The ROC analysis confirms the validity of the analysis with a high area under the curve (AUC) of 0.96 (supplementary figure S1e).

In addition, conducting ROC analysis using M.E scores and other computationally predicted scores such as CADD or FATHMM and comparing their performance would be informative.

Following the reviewer's suggestion, we have tested the predictions of the CSCAPE, FATHMM-XF and CADD tools in the neutral group 1 versus the effective group 2 in our screen in a ROC analysis. All three scores show only minor predictive power for this particular type of mutations with AUC values of 0.53, 0.54 and 0.52, respectively (supplementary figure S2b).

3. The evidence presented in the current manuscript is not sufficient to show whether the M.E. score obtained from the reporter system directly reflects the extent of protein degradation within cells. The authors evaluated protein expression only for three mutations observed in the PTEN gene and for C-terminal extensions in Figure 4a,b. However, this analysis includes only a few mutations that emerged as hits and does not include mutations with various ranges of M.E. scores as controls. I suggest that the authors include several additional controls, not just one PTEN gene, to confirm the correlation between the M.E. score obtained from high throughput screening and the extent of protein expression degradation.

To further validate the classification into neutral (group 1) and effective (group 2) extensions based on their enrichment in the screen, we have selected three genes from group 1 (three mutations: ACO2 *>G, GATA1 *>S, PRKCH *>Y) and three tumor suppressor genes from group 2 (five mutations: B2M *>Q, CDKN1B *>L and *>S, MLH1 *>K and *>Y) for further analysis. We have cloned the coding sequences and the 3' untranslated regions up to the next in-frame stop codon of these six genes and generated expression constructs for the six wildtype sequences (intact stop codon) and the eight nonstop mutant sequences. These constructs have been transfected and the protein expression levels have been determined using western blotting. In all eight cases, the western blot analysis of the coding sequence with the respective extension confirms the results from the functional report screen: for the

five mutations of group 2, there is a significant decrease in protein abundance, which is not seen for any of the three mutations from group 1 (figure 3f-m, supplementary figure S3a-h).

4. The authors suggested several features that determine nonstop-mediated loss of expression, such as length of the neutral extension, predicted hydrophobicity, composition of hydrophobic amino acids, existence of glycine at the last position, etc. Could the authors perform feature importance analysis (i.e. SHAP value) on various features suggested from the high-throughput analysis? This analysis could serve as important evidence for predicting the pathogenicity of nonstop mutations in the future.

Following the reviewer's suggestion, we have compared the median enrichment as well as the Spearman correlation with the enrichment level for 28 parameters (supplementary figure S5a-h).

For the hydrophobicity based on the Miyazawa scale or on the Kyte-Doolittle scale as well as for the extension length, the Spearman correlation to the enrichment in the functional is high and strongly significant (supplementary figure S5a,b). In contrast, the correlation coefficients for the twenty individual amino acids are all much lower (supplementary figure S5c). Comparing the median enrichment between extensions with components of known degrons like C-terminal glycine (G), an arginine at the last-but-third position (R at -3) also showed only weaker differences and for a C-terminal di-glutamic acid (EE), a C-terminal glutamine (Q) or a valine at the last-but-second position (V at -2), there are no significant differences in enrichment (supplementary figure 5d-h). Thus, out of these tested properties, the hydrophobicity and length of the extension show the strongest correlation with the median enrichment in the screen.

REVIEWERS' COMMENTS

Reviewer #1 (Remarks to the Author):

All of my concerns have been addressed.

Reviewer #2 (Remarks to the Author):

Overall, the authors have significantly improved the manuscript. Specifically, I asked for additional experiments to establish the associations between M.E. scores and the actual depletion of protein expression in several established control nonstop mutations. They have addressed my queries well and confirmed the association. I believe that the current manuscript does not require additional experiments and is suitable for publication.

I have only a few minor comments:

1. In line 119, "Using fluorescence assisted cell sorting (FACS)" should be "fluorescence-activated cell sorting (FACS)."
2. In Figure 1c, the authors should spell out BR (presumably biological replicate?) in full in the main text or figure legend.

Response Letter

First, we would like to thank the reviewers and the editors for positively evaluating our manuscript. Here, we provide a point-by-point response letter to the two latest decisions as requested.

Reviewer comments - editor decision 08.08.2024

Reviewer #1 (Remarks to the Author):

All of my concerns have been addressed.

Reviewer #2 (Remarks to the Author):

Overall, the authors have significantly improved the manuscript. Specifically, I asked for additional experiments to establish the associations between M.E. scores and the actual depletion of protein expression in several established control nonstop mutations. They have addressed my queries well and confirmed the association. I believe that the current manuscript does not require additional experiments and is suitable for publication.

I have only a few minor comments:

1. In line 119, "Using fluorescence assisted cell sorting (FACS)" should be "fluorescence-activated cell sorting (FACS)."

We have changed the text accordingly.

2. In Figure 1c, the authors should spell out BR (presumably biological replicate?) in full in the main text or figure legend.

We have changed the text of the figure legend 1c accordingly.